# Volumetric Modular Construction Risks: A Comprehensive Review and Digital-Technology-Coupled Circular Mitigation Strategies

Ayaz Ahmad Khan [1,*], Rongrong Yu [1], Tingting Liu [2], Ning Gu [1] and James Walsh [1]

[1] Australian Research Centre for Interactive and Virtual Environments (IVE), UniSA Creative, University of South Australia, Adelaide, SA 5000, Australia; rongrong.yu@unisa.edu.au (R.Y.); ning.gu@unisa.edu.au (N.G.); james.walsh@unisa.edu.au (J.W.)
[2] Cities Research Institute, School of Engineering and Built Environment, Griffith University, Southport, QLD 4222, Australia; tingting.liu@griffith.edu.au
* Correspondence: ayaz.khan@mymail.unisa.edu.au

**Abstract:** Volumetric modular construction (VMC) has considerable benefits in providing better cost, time, quality, productivity, and sustainability performance. However, the adoption is low, owing to various associated risks. This study aims to identify VMC critical risk factors (CRFs) in project stages and project attributes by conducting a systematic literature review of 91 articles. In the project stages, 31 CRFs were identified based on frequency analyses, and divided into four stages: design and planning, offsite manufacturing, transportation and logistics, and onsite assembly. The project attribute risks were divided into the following categories: implementation and schedule, supply chain and financial, safety and ergonomic, and civil and structural. Overall, design and planning risks were critical, as they are crucial in maintaining the project's upstream and downstream flow. Finally, the study proposed a mitigation framework for using digital technology-based circular strategies to overcome VMC risks. The framework includes disruptive and emergent digital technologies aiming to mitigate risks in VMC, keeping circularity in action. In terms of theoretical contributions, this research delivers a CRF register and categorical division for professionals to better understand the landscape of VMC risks. In terms of practical contributions, the study guides the practitioners towards strategies to overcome the pertinent risks.

**Keywords:** volumetric modular construction (VMC); project stage risks; project attribute risks; systematic literature review; digital technology; circular economy

## 1. Introduction and Background

The COVID-19 pandemic had significant detrimental impacts on the construction industry [1]. For example, the Australian construction industry, which contributes to around 9% of the country's GDP, has lost approximately USD 5 billion due to this pandemic [2]. Imposed lockdowns in many countries and new social distancing norms have resulted in a trend toward an increasing number of virtual inspections and fewer workers physically present on job sites [3]. Construction projects faced significant disruptions in the operations and supply chain, resulting in financial losses [4]. The abrupt changes caused by the pandemic have altered the way the architecture, engineering and construction (AEC) industries operate [5]. There is a solid need to transform traditional ways of construction and deliver the projects more effectively, efficiently, and safely. Volumetric modular construction (VMC) provides a viable option to achieve this goal. VMC integrates the concepts of modularity, modularization, lean manufacturing and production, design for excellence (DfX), and design for manufacture and assembly (DfMA), to deliver modules with fittings, fixtures and furnishing [6]. VMC emerges from the concepts of modularity, modularization, and

design for manufacture and assembly (DfMA) [7,8]. Modularity is defined as the phenomenon where different product components are made autonomously, yet have sufficient compatibility to be combined to develop an integrated system utilizing equivalent design specifications [9]. Constructing and deconstructing modules to accomplish varied results is a valuable outcome of modularity procedures. Comparably, modularization is an organized disintegration with rules, specifications, and boundaries to divide a building (system) into modules (components) [10]. DfMA is an approach that combines the methodologies of design for manufacture (DfM) and design for assembly (DfA), for an easy fabrication of modules and assembly process [6]. With its ease of DfM and DfA processes, DfMA aids in providing necessary speed, low cost, better quality, less time, high reliability, enhanced safety, and improved output for the VMC method [11].

Regarding material choices for VMC modules, concrete, steel, and timber are some of the commonly used along with combination of two or more [12]. For instance, the Hickory group, as one of the leading companies of VMC and other offsite construction in Australia, used steel modules to build the 44-level La Trobe Tower in Melbourne [13]. On the other hand, the International Convention Centre in Sydney was built using 4200 prefabricated concrete modules. Additionally, another leading company, named Lendlease, used cross-laminated timber (CLT) modules for building the International House in Sydney. Other examples of buildings in Australia that have utilized VMC include the Melbourne One 9 apartments, Peppers king square hotel in Perth, and the Adina apartment hotel in Sydney, among others [13,14]. These examples demonstrate the versatility of VMC and how it can be used with various materials to create efficient and sustainable buildings.

VMC is considered the process innovation in the competitive AEC industry. However, its uptake is still in the infant stage around the globe, with a few regions, such as Sweden, Japan, Hong Kong, and Singapore, being prominent players in VMC applications. The uptake in many other countries relatively lags behind due to the complex supply chain management, business models, and distinctive management involved in the VMC process [12]. For instance, vulnerabilities during the design stage in the VMC process include late design freeze, client change orders, ineffective shop drawings, and inadequacy in the design process [15]. Further, uncertainties at the factory stage in the VMC process include poor production plans, geometric variabilities in the modules, low capabilities of the manufacturers, and inefficient resource allocation [16].

Similarly, transport stage risks involve size and weight restrictions, delay in module delivery, absence of specific trunk routes and constraints in the road regulations [17]. Finally, there are risks, such as poor sequence planning, vertical and horizontal alignment defects, crane failures, and weather disruptions in the onsite assembly stage [18]. These potential risk events can derail the success of a VMC project, causing repercussions in cost, time, schedule, and quality. Nevertheless, according to the project management perspective, risks in construction are inevitable and should be thoroughly assessed, analyzed, and managed throughout the delivery chain of a project [19].

Previous studies have identified different risks in the VMC process around the world. For instance, in their study, Nabi and El-adaway [11] identified risks affecting the cost and schedule performance of VMC projects in the USA. In another study, they focused on identifying different disputes in the VMC and their causal relationships [20]. Similarly, Ekanayake et al. identified critical supply chain vulnerabilities in Hong Kong [17,21]. In Canada, Enshassi et al. [22] identified geometric variations in the VMC projects and proposed a framework for tolerance-based mitigation. Moreover, Gan et al. [23] focused on risks in VMC within the Chinese construction market, and identified various barriers to adopting VMC. Further, Gan et al. [24] also listed quality risk factors in VMC and identified causal relationships between them. Other studies highlighted the risks in the VMC process based on safety [25], schedule [26], and process barriers [27].

Even though previous studies have recognized the risks in VMC through different perspectives, no studies, to the best of our knowledge, have holistically addressed risks in VMC through the lens of project stages and project attributes [12]. Nonetheless, the unique

business model and complex supply chain activities of VMC spawn various risks [17]. The typical supply chain of the VMC comprises module design, material procurement, production process, warehouse accommodation, transportation logistics, buffer storage, and onsite assembly [21]. These complex stages involving a web of different stakeholders, with their specific purposes requiring a well-organized management procedure. To avoid distrust, uncertainty, and cynicism in the supply chain of VMC projects, it is imperative to highlight risks through the lens of project stages and project attributes [19]. As VMC projects involve multiple stakeholders, processes, and varied supply chain activities, identifying and categorizing the risks is imperative.

Consequently, this study pursues recognizing and classifying the risks in VMC through a systematic review of extant literature. Further, the most cited project stage critical risks from the theoretical and empirical studies are identified along with PARETO analysis of the project stage VMC risks. Finally, the study highlights the potential of digital technology (DT)-based circular strategies to address the risks in VMC. As a result, the following research questions need critical exploration:

1.　What are the risk-based VMC studies to date?
2.　What are the different project stages and project attributes of VMC risks?
3.　What is the potential of DT-based circular strategies for addressing VMC risks?

The answers to these crucial questions will develop a comprehensive understanding of VMC risks of interest to practitioners, academicians, researchers, industry professionals, and other stakeholders. The knowledge and information about VMC risks will allow for us to make better decisions in a categorical way, and optimize when the need for sustainable and circular methods, such as VMC, is at a high demand in the construction industry. The next section will reflect on the VMC background, followed by the research method in Section 3. The results and discussions will be discussed in Section 4, with conclusions in the last section.

## 2. Materials and Methods

This study follows the paradigms of a systematic literature review (SLR) as the research method to theoretically identify, categorize, and explain the risks in the VMC process. The SLR process is predominantly used in research studies to identify, theorize, and develop themes from a corpus of literature in the respective domain. Figure 1 shows an overall outline of the research design for the study.

The first step is to extract and evaluate the existing studies on VMC risks and retrieve the relevant articles. After retrieving articles, a mixed review method is used, which includes a systematic literature review and critical content analysis of the articles to examine the VMC risks involved during the project stage and the project attributes. The in-depth mixed review is utilized to understand the taxonomy of VMC risks and identify gaps and future directions for mitigating those risks. The following sub-sections will describe the details of the two steps.

### 2.1. Extraction and Evaluation of Relevant Articles

The Preferred Reporting Items for Systematic Reviews and Meta-Analyses (PRISMA) protocol was utilized to retrieve the papers related to risks in the VMC stages. The PRISMA protocol is a traditional method used in scientific studies, which can reduce the risk of bias and enhance the validity of the findings from the literature [28].

Regarding the database selection, the two most common search engines, Scopus and Web of Sciences (WoS), were used to retrieve the relevant papers. These databases are major databases for indexing the relevant literature and are used widely in construction domain studies [29]. Following the precedents of previous review studies in VMC risks [12,30], an extensive search string of keywords was developed. As such, the following search keyword string was used for initial articles searching in the Scopus and WoS databases:

TITLE ("volumetric modular construction" OR "modular construction" OR prefabrication OR prefabricated OR "offsite construction" OR "offsite manufacturing" OR "offsite

production" OR "modern method of construction" OR "industrialized construction" OR "industrialized building systems" OR "systems building" OR "modular integrated construction" OR "prefabricated prefinished volumetric construction") AND TITLE (risk* OR barrier* OR challenge* OR problem* OR obstacle* OR hindrance*).

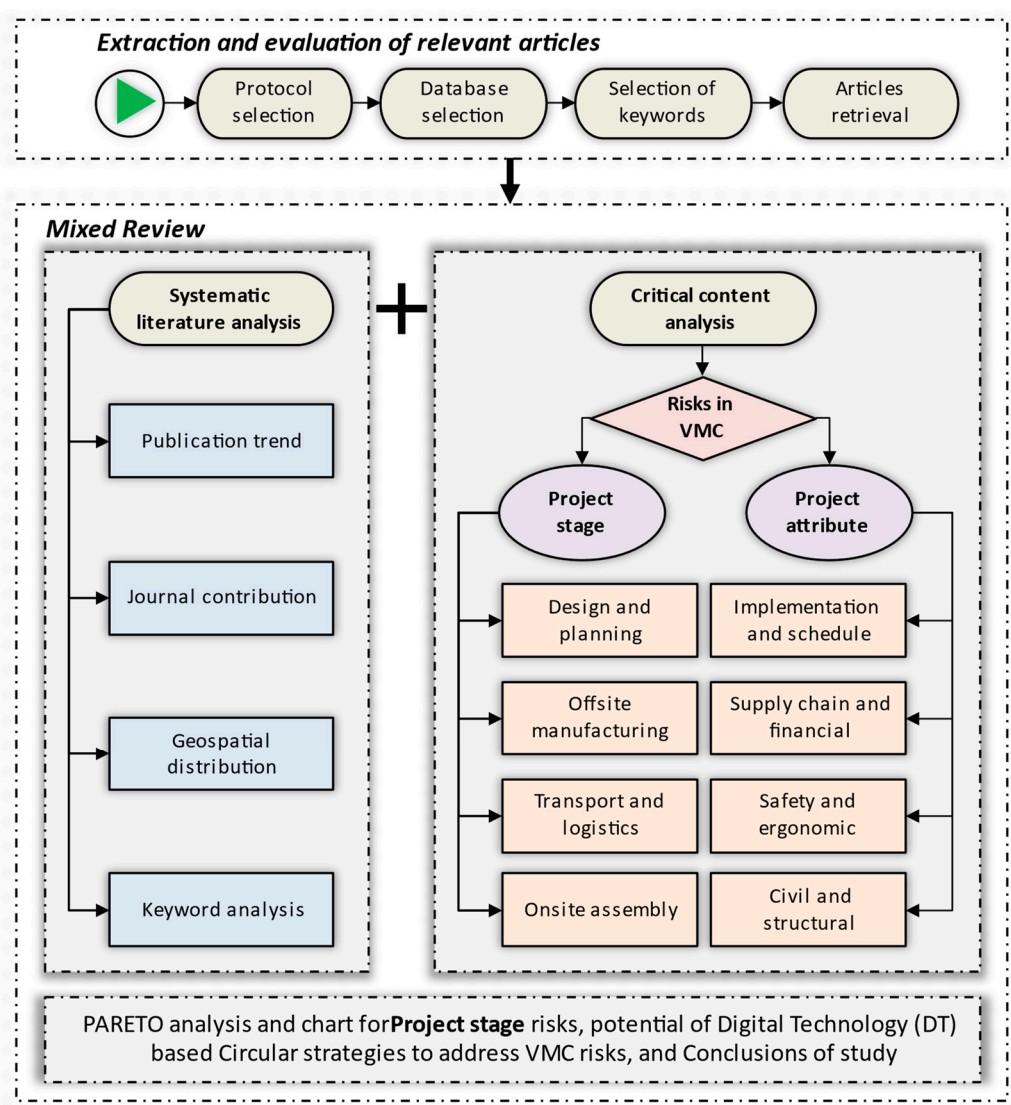

**Figure 1.** Overall research design for the study.

The search duration is limited to the last decade (1 January 2010–20 January 2023) to identify the recent focus on the risks in the VMC method. This is also because studies on VMC risks were primarily published during the last decades, since VMC is a predominantly new construction method [31]. The document types were restricted to articles and reviews, as they potentially deliver more superior knowledge than conference papers due to the relative more rigorous peer-review process. Similarly, the source type was confined to journals only, and finally, English was kept as the language to retrieve the papers. Next, manual screening of the articles was conducted based on a reading of the title, keywords, abstract, and conclusion (since the scope of the work and research objectives can be understood through its title, the keyword used, and abstract and conclusion reading) [28].

Following the rigorous process, a total number of 91 papers relevant to risks in the VMC stages were included in the study. The steps of the PRISMA protocol are described in Figure 2. The list of 91 papers is listed in Appendix A at the end of the paper.

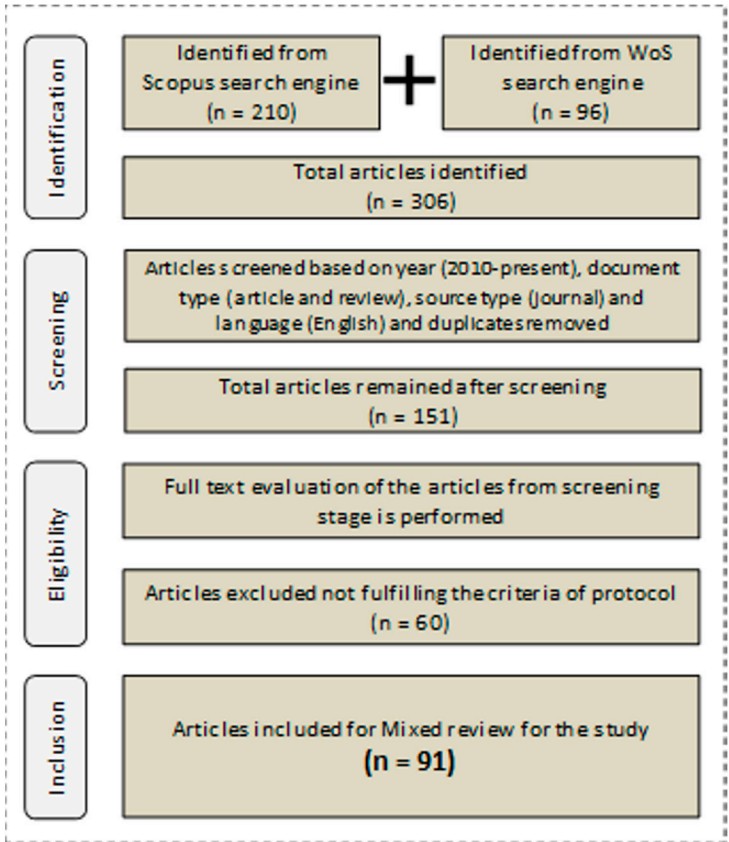

**Figure 2.** PRISMA protocol flow diagram.

*2.2. Mixed Review Process*

The mixed review process comprises systematic literature analysis and critical content analysis, as shown in Figure 1. This method can deliver a deeper understanding of the nature of retrieved articles from different perspectives, and extract relevant knowledge to present critical content of the existing studies [32]. The systematic literature analysis comprises publication trends, journal contribution, geospatial distribution, and keyword analysis of the retrieved articles.

The included 91 articles were used to extract critical content to explore the VMC risks, both the project stage and attribute risks. This study identified a comprehensive list of 67 project stage risks (Appendix B); among them, 31 critical risks factors (CRFs) were highlighted based on cumulative frequency analysis. These risks were divided into four categories of project stage risks. Moreover, PARETO analysis was conducted for the four individual project stage risks. PARETO analysis is a formal technique to explore the causes of a problem when many reasons are responsible for derailing a specific event or project. It works based on the assumption that roughly 80% of the influences come from 20% of the reasons [33]. The cumulative frequency is 100%, such that the "vital few" CRFs reflect 80% of the cumulative percentage of citation frequencies, and the "trivial many" CRFs occupy 20% of occurrences. As this review focused on frequency analysis to highlight the risks in VMC, PARETO analysis will be useful for prioritizing the CRFs. Regarding project attribute risks, a careful examination of 91 articles was conducted to synthesize the knowledge and division into four sub-categories. Finally, digital technology-based circular strategies were proposed to overcome VMC risks. The following sections present the results of the mixed review process.

## 3. Systematic Literature Analysis

### 3.1. Publication Trend over the Selected Period

The publication trend for a topic depicts its relevance and importance in the field among researchers and other stakeholders in the industry. The distribution of the reviewed articles (91) per year is shown in Figure 3. It is clear from the figure that an average of one to two articles focused on risks in VMC, from 2010 to 2016, except for 2014, which had four publications related to VMC risks. This reflects that the attention toward VMC risks was minimum during this time. This is in line with the review by Hosseini et al. [34], stating that dedication toward VMC and other OSC techniques has been slow during the initial half of the last decade due to insufficient knowledge and awareness in the construction industry.

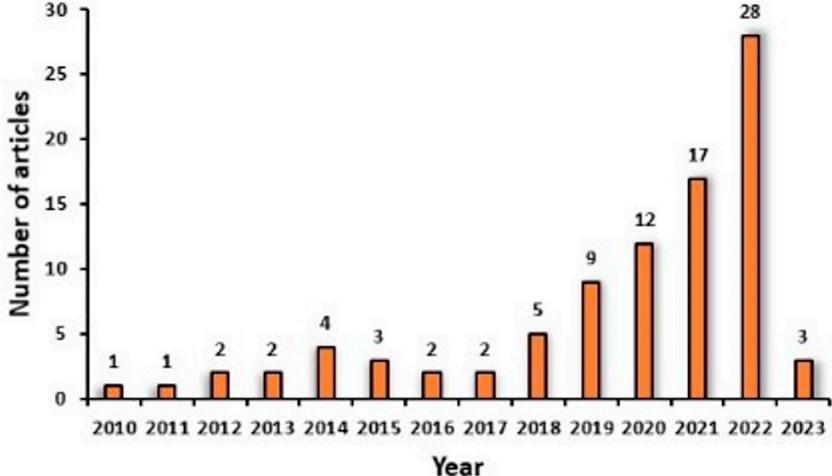

**Figure 3.** Publication trend over the selected time duration.

However, a rapid rise in articles was seen from 2017 to the present. This escalating trend shows the advancement in studies about risks associated with VMC and other OSC types, thus promoting the wide VMC utilization within the construction industry [7]. A significant rise in the number of articles (24) was seen in 2021. We can infer that the COVID-19 pandemic may have escalated the usage of VMC owing to its fast construction time, less workforce required, and sustainable measures [31]. The post-pandemic world will see a significant application of VMC and other OSC types [1] that reinforces the need for this study to highlight the CRFs at different stages of VMC.

### 3.2. Key Journals for Publishing the Topic

The analysis of journal distribution highlights the impact and quality of the articles studied. It also provides a reference for the researchers to select the specific journal related to VMC risks. Table 1 below the distribution of 91 articles in the peer-reviewed journals included in this study.

The articles were published widely in 25 journals; among them, 12 journals published at least two articles each. The rest of the 13 journals have published one article on VMC risks. With 20 articles (22.9%), JCP is among the top contributing journals. Due to its characteristics, such as its sustainable, environmentally friendly and energy-efficient construction methods, the number of VMC-related articles in JCP is logical, since the journal's focus is well aligned to VMC-related studies [12]. Furthermore, the articles published in this study in the following four journals, namely ECAM, JME, IJCM, and JCEM, are 12, 8, 8, and 6, respectively. Those four journals focus on the management aspect of the AEC industry.

**Table 1.** Journal distribution of the study articles.

| Journal Name | No. of Articles |
|---|---|
| *Journal of Cleaner Production* (JCP) | 22 |
| *Engineering, Construction and Architecture Management* (ECAM) | 12 |
| *Journal of Management in Engineering* (JME) | 8 |
| *International Journal of Construction Management* (IJCM) | 8 |
| *Journal of Construction, Engineering and Management* (JCEM) | 7 |
| *Sustainability* | 5 |
| *Automation in Construction* (AiC) | 5 |
| *Applied Sciences* (AS) | 3 |
| *Construction Management and Economics* (CME) | 2 |
| *Buildings* | 2 |
| *Building and Environment* (BE) | 2 |
| *Journal of Civil Engineering and Management* (JoCEM) | 2 |
| *Journal of Facilities Management* (JFM) | 1 |
| *Journal of Asian Architecture and Building Engineering* (JAABE) | 1 |
| *International Journal of Injury Control and Safety Promotion* (IJICSP) | 1 |
| *International Journal of Construction Education and Research* (IJCER) | 1 |
| *Habitat International* (HI) | 1 |
| *Ergonomics* | 1 |
| *Computers in Industry* (CI) | 1 |
| *Canadian Journal of Civil Engineering* (CJCE) | 1 |
| *Applied Ergonomics* (AE) | 1 |
| *Journal of Computing in Civil Engineering* (JCCE) | 1 |
| *Construction Innovation* (CIn) | 1 |
| *Frontiers of Engineering Management* (FEM) | 1 |
| *KSCE Journal of Civil Engineering* (KSCE-JCE) | 1 |

The involvement of various stakeholders and different supply chain stages in VMC requires novel management techniques and strategies to oversee and prevent numerous potential risks, which may explain the inclination of researchers toward these journals. Additionally, the sustainable features and facilitation of automation by the VMC method studies tend to be published in journals such as *Sustainability* and *AiC*. A few influential journals getting the attention of VMC researchers are CI, JCCE, and CIn, as these journals focus on utilizing digital technologies with construction techniques. Recently, with the advent of the Industrial Revolution 4.0 (IR-4.0), the rise of utilizing digital technologies to solve construction issues has gained momentum [35]. A few studies have also used digital technologies integrated with VMC to unravel new possibilities. Finally, journals such as AE, IJICSP, and *Ergonomics* have published articles related to health and safety in VMC, such as worker injury related to the low back [36], worker safety [25], and work-related musculoskeletal disorders (WMSDs) [37].

*3.3. Geographical Distribution of the Selected Studies*

The geographical distribution of the articles reflects the interest and advancement of VMC-related literature in different regions. As Figure 4 suggests, Mainland China has contributed 31 articles on VMC risks. With a developing economy, Mainland China has clear visions, policies, and strategies to apply VMC, owing to its large population density and share in the carbon footprint [23]. Similarly, the Hong Kong region shares the contribution of 21 articles.

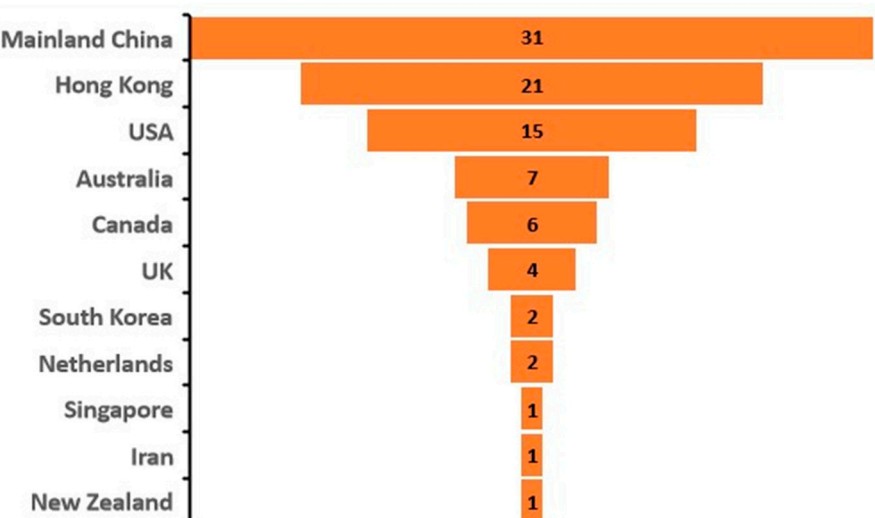

**Figure 4.** Geographical distribution of the studies articles.

With a developing economy, Hong Kong also possesses a strategic roadmap for VMC implementation [38]. The share of other regions focused on VMC risks is the USA (14), Australia (7), Canada (6), the UK (4), South Korea (2), and the Netherlands (2). Further, one article from other developing Asian regions, such as Pakistan and Iran, shows their initial steps toward cleaner and sustainable construction methods. Finally, it is understandable to see no contribution from African and South American countries, as they lack advanced application and utilization of VMC methods.

*3.4. Keyword Analysis of the Reviewed Articles*

The cluster of keywords and their linkage reflects the focus interests of the studies, along with promising integration opportunities with different keywords [39]. Figure 5 shows the keyword mapping of the 91 selected studies using the science mapping software, Vos-Viewer. As seen in Figure 5, different clusters of keywords are identified from the mapping of the studies included.

The analysis revealed seven prominent clusters in which the risk studies are classified. Cluster 1 includes studies focused on risks due to the pre-design and design stage disruptions, and is the largest cluster. It is evident from this cluster that design stage risks are significant for the VMC method, as decisions taken at this stage affects the downstream stages of the VMC project. Further, cluster 2 is related to studies mentioning stakeholder and contracting issues. Next, cluster 3 emphasizes the building information modeling (BIM)-integrated risks at different stages. Several studies highlight that the risks occurred due to the utilization of BIM ideology at different phases of the VMC project.

Moreover, cluster 4 sheds light on VMC risks in housing and residential buildings. VMC is deemed to be appropriate for housing purposes; however, researchers identified bespoke risks and uncertainties in the VMC implementation for culminating housing shortage problems. The supply chain vulnerabilities were the focus of cluster 5 and barriers pertaining to sustainability issues were discussed in cluster 6. Lastly, cluster 7 underlines the safety and health-related risks in the VMC process. These clusters aid in identifying the project stage and project attribute risks in a more holistic way.

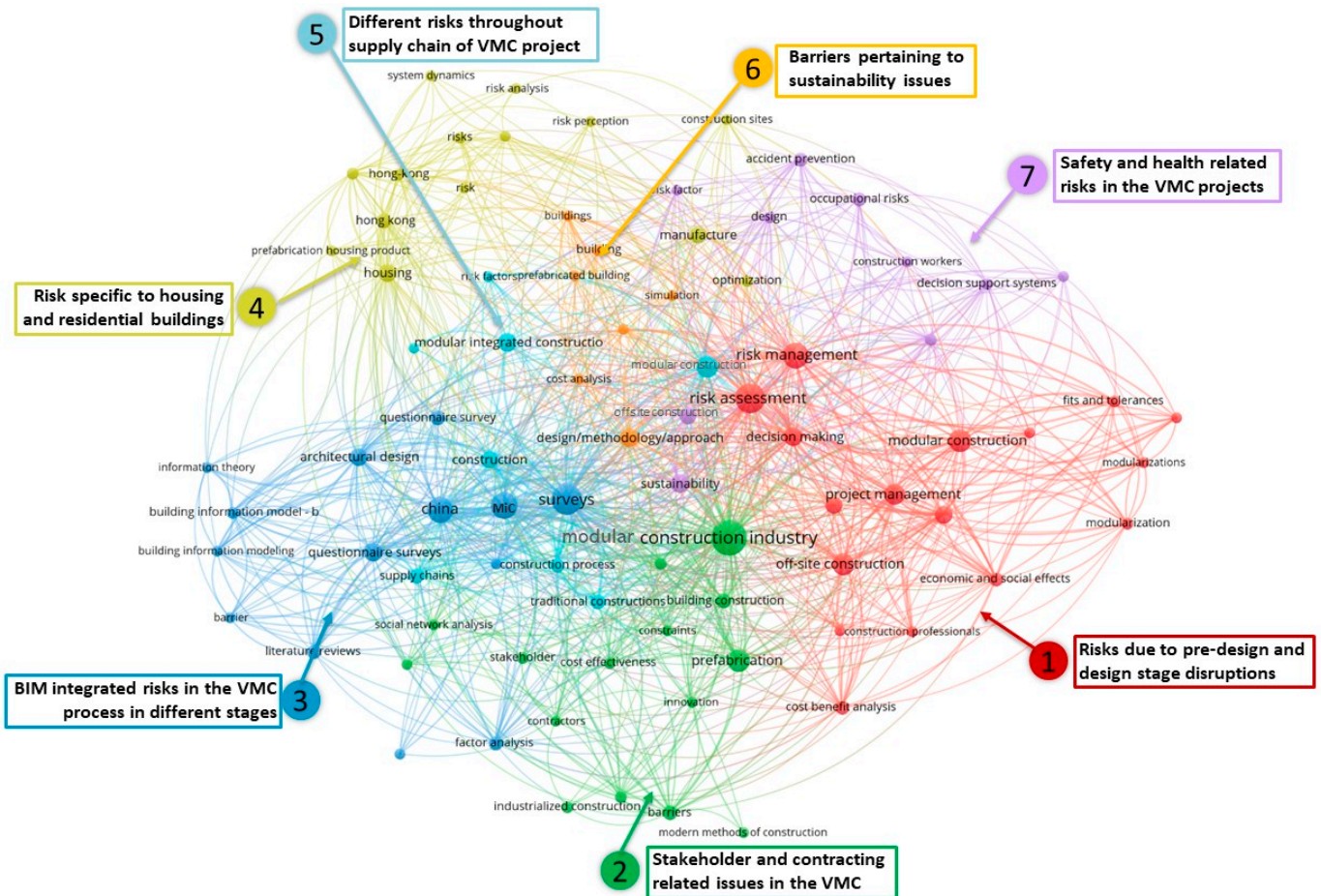

**Figure 5.** Keyword mapping of the reviewed articles.

## 4. Results and Discussions of Critical Content Analysis

### 4.1. Exploring Project Stage CRFs

The cumulative score (CS) based on the frequency analysis is utilized to explore the project stage risks. This method of getting critical risks through CS has been implemented in previous studies in the construction domain. It effectively analyzes the most significant inhibitors in a specific domain [12]. The 67 project stage risks were divided into categories based on four essential VMC stages, namely "design and planning" (DP), "offsite manufacturing" (OM), "transportation and logistics" (TL), and "onsite assembly" (OA). The four categories had 17, 18, 17, and 16 risks, respectively. Frequency and weight analysis were then conducted to identify the critical risks. The weight of the risks was calculated based on the following equations:

$$\text{Total weight of a risk (TW)} = (W_h \times 5) + (W_m \times 3) + (W_l \times 1) \qquad (1)$$

$$\text{Score of risk (TS)} = F + TW \qquad (2)$$

$$\text{Relative Score (RS)} = \text{Score of risk (TS)}/\text{Category Score} \qquad (3)$$

where $F$ represents the frequency of the risk occurrence in the retrieved articles, $W_h$ implies the number of papers having high weight (5), $W_m$ implies the number of papers having medium weight (3), and $W_l$ implies the number of papers having low weight (1) for a risk [29]. The high weight implies that the study directly mentioned the risk factor, the medium weight implies that the study indirectly mentioned the risk factor, and the low

weight relates to the studies implicitly highlighting the risk factor [29]. Utilizing these frequencies, the total weight (*TW*) and a score of the risk (*TS*) are calculated. Further, the relative score (*RS*) is calculated using the abovementioned equation. In Equation (3), the category score is a sum of TW for every four categories; for instance, 17 risks in the design and planning stage have a category score of 599, which adds all the TWs of that category. Finally, the risk CSs are calculated, keeping 70% as the threshold (Table 2), and 31 key risks were identified.

**Table 2.** Critical risk factors in VMC project stages.

| Group | Risks | Code | F | Weight | | | TW | TS | RS | CS | Ref. |
|---|---|---|---|---|---|---|---|---|---|---|---|
| | | | | $W_h$ | $W_m$ | $W_l$ | | | | | |
| Design and planning | Change order/design freeze issues from the clients | DP1 | 17 | 8 | 5 | 4 | 59 | 76 | 0.13 | 0.13 | [12,19,20,30,40–51] |
| | Complexity in the modular designs/rigid geometry | DP2 | 15 | 7 | 5 | 3 | 53 | 68 | 0.11 | 0.24 | [12,19,20,30,40,42,44–51] |
| | Design changes and defects in the modules size | DP3 | 16 | 8 | 4 | 3 | 55 | 71 | 0.12 | 0.36 | [12,19,20,30,40,42,44–52] |
| | Coordination problem between project participants | DP4 | 13 | 6 | 4 | 3 | 45 | 58 | 0.10 | 0.46 | [20,27,30,40,46–51,53,54] |
| | Shop drawing management problems/unclarity | DP5 | 10 | 5 | 3 | 2 | 36 | 46 | 0.08 | 0.53 | [19,27,40,49,55–60] |
| | Lack of BIM and visualization techniques in the design | DP6 | 8 | 4 | 2 | 2 | 28 | 36 | 0.06 | 0.59 | [22,24,61–66] |
| | Inadequate codes and standards of VMC locally | DP7 | 11 | 6 | 3 | 2 | 41 | 52 | 0.09 | 0.68 | [19,24,25,40,52,61,67–71] |
| Offsite Manufacturing | Poor understanding of process plans/system failure | OM1 | 16 | 9 | 4 | 3 | 60 | 76 | 0.11 | 0.11 | [16,17,20,21,27,30,45,47,50,53,54,59,66,72–74] |
| | Noise, fume, and toxic compound exposure at the plant | OM2 | 10 | 5 | 4 | 1 | 38 | 48 | 0.07 | 0.18 | [16,25,36,37,46,75–79] |
| | Conflicts in geometry of modules from the design phase | OM3 | 15 | 8 | 5 | 2 | 57 | 72 | 0.10 | 0.28 | [12,16,17,22,27,54,60,62,66,70,71,74,80–82] |
| | Inadequate inventory control and shortage of material | OM4 | 13 | 7 | 4 | 2 | 49 | 62 | 0.09 | 0.37 | [16,17,21,36,40,65,70,83–88] |
| | Poor/inexperienced labor and resource allocation | OM5 | 12 | 6 | 4 | 2 | 44 | 56 | 0.08 | 0.45 | [17,22,44,63,65,66,70,75,76,78,87,88] |
| | Lack of modern equipment for the lifting process at the plant | OM6 | 11 | 5 | 3 | 3 | 37 | 48 | 0.07 | 0.52 | [16,17,36,37,54,62,76,89–92] |
| | Defects due to welding process/geometric variations | OM7 | 10 | 6 | 2 | 2 | 38 | 48 | 0.07 | 0.58 | [12,22,51,62,65,71,76,80,89,91] |
| | Inadequacy in the weather proofing and space usage | OM8 | 11 | 6 | 3 | 2 | 41 | 52 | 0.07 | 0.66 | [16,17,42,45,54,62,65,86,91,92] |
| Transportation and Logistics | Delay in delivery/poor scheduling of modules | TL1 | 13 | 7 | 4 | 2 | 49 | 62 | 0.11 | 0.11 | [17,21,26,42,47,54,58,59,65,72,81,93,94] |
| | Defects by damage/flexing/warping and manual handling | TL2 | 10 | 5 | 4 | 1 | 38 | 48 | 0.09 | 0.20 | [17,22,40,56,62,63,82,88,89,95] |
| | Size and weight restrictions in transportation | TL3 | 12 | 6 | 4 | 2 | 44 | 56 | 0.10 | 0.30 | [17,21,26,45,46,54,59,65,71,83,96,97] |
| | Restrictions of rules, regulations, and transport vehicles | TL4 | 11 | 5 | 3 | 3 | 37 | 48 | 0.09 | 0.38 | [17,26,45,59,63,65,71,83,87,96,97] |
| | Early arrival and wrong delivery of modules on site | TL5 | 10 | 5 | 3 | 2 | 36 | 46 | 0.08 | 0.47 | [12,26,45,46,59,65,71,83,96,97] |
| | Poor marking/tagging and improper buffer space onsite | TL6 | 11 | 6 | 3 | 2 | 41 | 52 | 0.09 | 0.56 | [27,43,45,46,53,58,59,66,91,93,98] |
| | Distance issues and taxes incurred between the plant and the site | TL7 | 8 | 4 | 2 | 2 | 28 | 36 | 0.06 | 0.62 | [26,45,59,65,71,83,96,97] |
| | Misplacement of the modules in the warehouses causes delay | TL8 | 8 | 4 | 3 | 1 | 30 | 38 | 0.07 | 0.69 | [16,21,26,55,58,60,93,94] |

**Table 2.** *Cont.*

| Group | Risks | Code | F | Weight | | | TW | TS | RS | CS | Ref. |
|---|---|---|---|---|---|---|---|---|---|---|---|
| | | | | $W_h$ | $W_m$ | $W_l$ | | | | | |
| Onsite Assembly | Inefficient lift path/layout planning of the crane(s) and scheduling/sequencing of the modules | OA1 | 16 | 8 | 5 | 3 | 58 | 74 | 0.11 | 0.11 | [19,26,27,30,40,47,47,51,54,58,59,72,91,92,99,100] |
| | Poor stability/blind lifting, frequent breakdown of the crane and change in rigging direction | OA2 | 15 | 7 | 4 | 4 | 51 | 66 | 0.10 | 0.20 | [12,21,22,25,26,54,55,58,60,65,70,88,94,101,102] |
| | Break of the cable crane/jib falling and extra load on the crane | OA3 | 13 | 6 | 3 | 4 | 43 | 56 | 0.08 | 0.29 | [17,22,46,65,81,83,88,91,92,94,95,99,102] |
| | Unsafe acts/conditions and errors in installations onsite | OA4 | 15 | 8 | 4 | 3 | 55 | 70 | 0.10 | 0.39 | [25,30,53,70,71,76,78,87,89–91,93,94,99,103] |
| | Poor verification due to inadequate tagging/inefficient welding | OA5 | 12 | 6 | 4 | 2 | 44 | 56 | 0.08 | 0.47 | [22,25,46,51,62,65,71,76,78,80,89,91] |
| | Manual lifting/unwrapping/lining/ unhooking and screwing | OA6 | 10 | 5 | 3 | 2 | 36 | 46 | 0.07 | 0.54 | [16,22,36,43,44,80,82,83,87,91] |
| | Variabilities in geometry/ dimensions and poor alignment of the modules | OA7 | 13 | 7 | 4 | 2 | 49 | 62 | 0.09 | 0.63 | [16,45,48,53,62–65,71,79,80,85,99] |
| | Wind/weather and near-environment disruptions at the site | OA8 | 8 | 4 | 2 | 2 | 28 | 36 | 0.05 | 0.68 | [17,21,25,40,53,58,88,94] |

*4.2. Project Stage Risks*

4.2.1. Design and Planning

Although design and planning are the first stages in developing a VMC project, the ideal process of VMC should begin with the manufacturer finalizing the materials and components [61]. However, the process is seldom ideal, and in most cases, the manufacturer starts with the architect's model without providing necessary information. The lack of communication causes risks at this initial stage, as the accurate details needed by the manufacturer in terms of screws, bolts, welds, and kits of various parts are not standardized as per the manufacturer's capabilities. Figure 6 shows the PARETO chart for design and planning risks. The CRFs from DP-1 to DP-4 are vital and responsible for 80% of the effect on the design and planning stage. As such, these CRFs require additional focus by the project teams during the implementation of the VMC project.

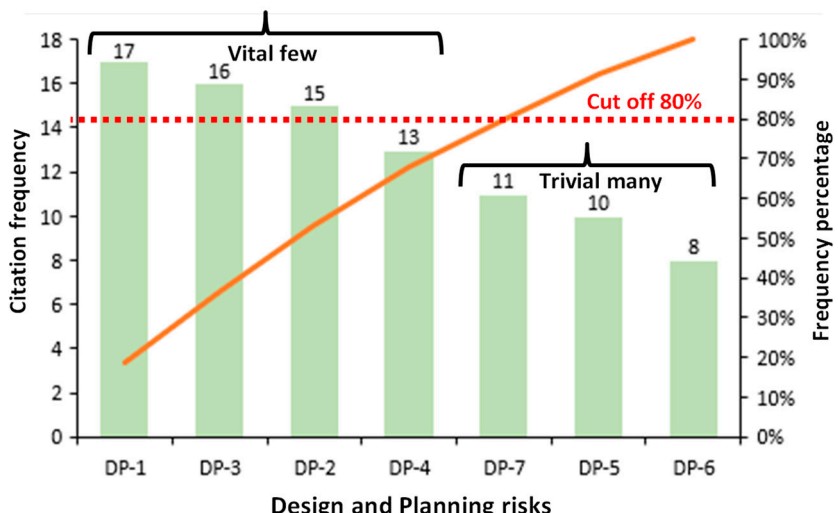

**Figure 6.** PARETO chart for design and planning risks in VMC.

Further, the architect needs to understand that modular construction is more inclined toward the manufacturing side to achieve a successful project, and is further exacerbated by the client's discrepancy in finalizing the design (DP-1) [19,30]. Regarding this, the integrated design process, such as DfMA, is necessary for addition to the application of concepts of modularity and design for excellence (DfX) to manage the complexity involved in modular design and geometry (DP-2) [11]. DfX includes a broad spectrum of strategies necessary

in the initial stage of a VMC project. A few significant DfX strategies are designed for assembly, disassembly, adaptability, buildability, constructability, deconstruction, deploy ability, and flexibility [104]. Other DfX strategies that are used at the end-of-life cycle of a project are designed for reuse, recycle, reverse logistics, transformability, and waste minimization [104]. Further, the BIM-authoring software utilized by architects differs from manufacturers in most cases. Thus, the level of detail (LOD) provided by the architect is not conceived adequately by the party responsible for module development (DP-6) [61]. These uncertainties and communication gaps cause various risks at the design stage, thus causing disruptions in the downstream stages of VMC.

For instance, one of the major causes is the inability to provide adequate working drawings suitable to produce the modules (DP-5) [27]. Further, the absence of manufacturing experts' inputs at the design stage results in a poor development of shop drawings, thus resulting in downstream risks caused by the design. Moreover, late or no involvement of fabricators, factory contractors, and suppliers becomes the recipe for inadequate design information and communication (DP-4) [47,51]. The snowballing effects of preliminary design drawings may further cause cascading effects at the later stages of the VMC project. Any late changes then become tricky and expensive to implement, causing a delay in the overall supply chain at the consequent stages (DP-3) [21,54].

In addition, the ineffective design information further extends the lead time, resource allocation, and production processes. In addition, the lack of DfMA principles and proper structural efficacies cause horizontal and lateral vulnerabilities in the integrity of the structure design [19]. These can lead to dimensional and geometric variabilities in the development of modules, ultimately resulting in the over-production of error-laden modules [71]. A good practice to avoid such design susceptibilities is the early involvement of manufacturing experts to minimize further changes and freeze the design information promptly [12]. Further, specific codes and standards are necessary to streamline the module design process (DP-7) [24]. Therefore, the effective management of DP risks is significant to reduce the cascading issues in the downstream stages of the VMC project.

### 4.2.2. Offsite Manufacturing

The risks in the OM stage are caused by the disruption in the decisions, processes, events, actions, and measures taken during the factory production of the modules [16]. These traits leverage the capabilities of factory manufacturers to deliver high-quality VMC modules, thus aligning with project-specific requirements and overall gains from the VMC method. Markedly, the manufacturing stage acts as a conduit bridge between the other upstream and downstream stages throughout the VMC project [90]. Therefore, it is necessary to minimize the risks at this stage and achieve better optimization of the three pillars of cost, time, and quality, along with attaining the energy efficiency and sustainability yields of VMC [16]. The inconsistent design information, late design freeze, and insufficient working drawings from the design stage are suggested to be recipes for manufacturing risks and can extend the production time of the modules (OM-3) [61]. Figure 7 shows the PARETO chart for offsite manufacturing risks. The five CRFs, namely OM-1, OM-3, OM-4, OM-5, and OM-6, are vital and responsible for 80% of the effect on the offsite manufacturing stage. Therefore, these CRFs need extra attention from factory manufacturers during the module production process.

Further, the decisions of manufacturer selection can lead to discrepancies such as the overproduction of modules, deviations in production and operation rates, poor module production and ineffective resource utilization [59]. However, the limited manufacturing unit capacity is due to a low number of VMC projects, which restricts the high investment in developing efficient manufacturing yards. In many countries, manufacturing setups' low capabilities are significantly due to less interest in VMC methods, which rely heavily on the government and official bodies to promote VMC projects [16].

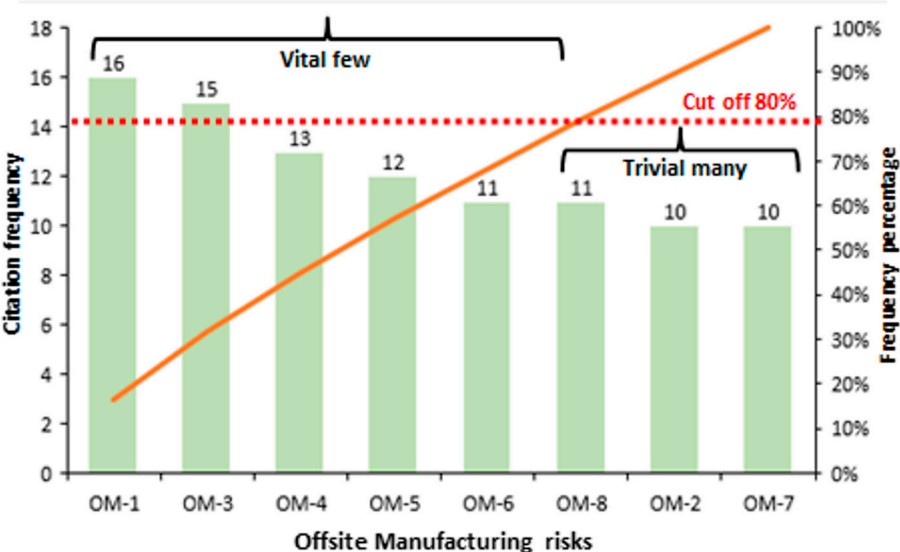

**Figure 7.** PARETO chart for offsite manufacturing risks in VMC.

Thus, many manufacturing facilities have a low competency in the skills of workers (OM-5) and automation techniques, and fail to achieve immediate process plans for the production of modules (OM-1) [78]. In addition, low competency results in meagre optimization of materials and resource allocation (OM-5) to achieve effective production line measures, causing subdued resource utilization time, and thus the overall dwindling lead time for module manufacturing [105]. Additionally, the amount of waste generated due to inefficient manufacturing in modular factories is a concern, further escalating in generating noise, fumes, and toxic compounds in the factory yard(s) (OM-2) [46]. Moreover, lacking initial collaboration with the architect and other communication gaps led to inadequate inventory control and the factory's material shortage (OM-4) [12].

The other notable risks that can hinder the manufacturing process are related to ancillary items or tasks that aid in the smooth production of modules in the factory. The factory processes heavily rely on manual operations that constrain the production speed and productivity [78]. A significant lack of automated lifting at the plant disrupts the factory logistics process and threatens workers that rely on manual operations and tasks (OM-6) [106]. Automated guided vehicles (AGVs) are a potential solution for automating manual tasks in factory production [107]. The AGVs aid in lifting, transporting, and overall logistics when handling heavy modules in the factory [107]. However, due to the low capabilities of manufacturing plants and the high cost of AGVs, the implementation is difficult to apply in the VMC production factories widely.

Furthermore, the inefficient welding process of the modules creates geometric variations in the modules (OM-7) [22]. The lack of information from the designing phase relating module joinery can create issues in the welding process that ultimately results in deformation, cracks, the inclusion of slag, and incomplete penetration [46]. Additionally, less experience in the manufacturing unit further elevates this issue and can result in geometric intolerances in the modules. The last aspect is the weatherproofing of the modules to maximize life and performance (OM-8). Conventional weatherproofing techniques seldom work in VMC methods [61], thus leaving risks of properly sealing joints and filing. The necessary action in this regard needs to be taken at the design stage, where novel solutions, such as providing adequate gasketing and sliding seal options, should be carefully included in the shop drawings of the modules [61].

The success determinants of the OM stage are early collaboration during the design stage, before the final development of working drawings. Further, best practices, such as lean manufacturing and six sigma methodologies, should be implemented [48]. The training of workers should be provided to enhance the skills and workmanship to complete

the production tasks. Moreover, sufficient time should be allocated to ensure the adherence of developed modules by local registration certification bodies and any maneuverability in the different production value or non-value tasks [16].

### 4.2.3. Transportation and Logistics

The risks at the TL stage are critical, as this phase acts as an essential connection between the production and assembly processes. To ensure an optimal schedule of onsite works, it is imperative to have smooth transport of modules, and further requires planned logistics management. There is a considerable number of symbiotic risks that can occur at this stage. Initially, due to the inefficiencies in the timely completion of the modules by the factory, a delay in the delivery is likely to happen [12]. This issue not only disturbs the schedule of the onsite works (TL-1), but is also responsible for the extra cost incurred in the transportation process [19]. Figure 8 shows the PARETO chart for transport and logistics risks. The five CRFs, namely TL-1, TL-3, TL-4, TL-6, and TL-2, are vital and responsible for 80% of the effect on the transport and logistics stage. Hence, these CRF needs must get additional concentration during the transport and logistics of the VMC modules.

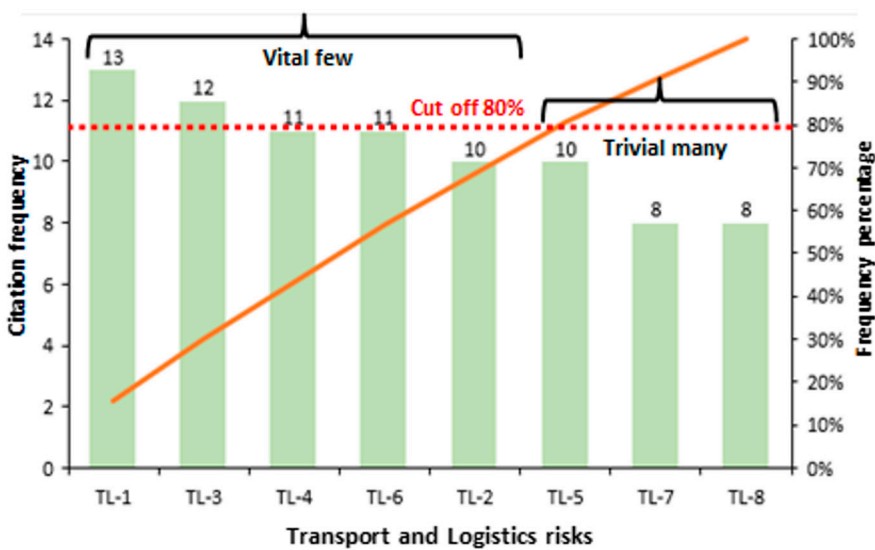

**Figure 8.** PARETO chart for transportation and logistics risks in VMC.

Further, it is reported that the risk of the wrong delivery of modules can occur on the site due to tighter schedules. In contrast, the early arrival of modules can sometimes happen, which can be a significant vulnerability if the adequate buffer or hedging space is unavailable onsite (TL-5) [12]. Moreover, instances of wrong module delivery have been reported in previous studies that reflect on the poor logistics planning in the transport of the modules. In this regard, just-in-time (JIT) delivery is suitable; however, the insufficiency and absence of onsite warehouses do not accommodate the JIT delivery method [48]. A few instances have been reported of the misplacement of modules or wrong delivery of modules in the warehouses and buffer spaces present onsite (TL-8) [21]. These unfortunate risks can arise due to the wrong handover of modules when they arrive onsite, further exacerbated due to improper tagging and marking of the modules (TL-6) [17].

The restriction In the size and weight of the modules is also a significant risk event, as it causes transportation issues and delays within the assembly stage (TL-3) [54]. The susceptibility of this risk event is mainly due to the cross-border import of the modules, where the rules, regulations, and standards differ depending on the country to which the modules are transported (TL-4) [12]. Due to this, interruptions such as port stoppage, custom clearance, and other symbiotic events, can occur [17]. In the case of local module manufacturing, the transportation of oversized and overweight modules is influenced due to the lack of specific trunk routes, narrow, dense urban environments, and inadequate



road conditions [74]. Furthermore, in the case of cross-border import of modules, the risk of damage, flexing, warping and manual handling can occur, which significantly affects the condition of the modules and, eventually, the quality of the project (TL-2) [46]. Further, instances of inclement weather and plum rain were also reported in previous studies as the cause of severe module defects during transportation [17]. These distance issues and taxes incurred during transportation unnecessarily add to the project costs and can be avoided with proper logistics planning (TL-7) [21].

Nevertheless, the transportation and production costs cover up to 60% of the project's total investment, and effective planning is required to manage the risks during these stages [17]. The use of a smart tagging system, JIT delivery, local manufacturing of modules, and proper logistics planning are necessary to reduce transportation risks, and hence avoid the late delivery of the project onsite [48]. Moreover, the local government must implement specific rules and regulations regarding the smooth transportation of modules.

### 4.2.4. Onsite Assembly

The success of a VMC project heavily relies on the OA tasks, marking the end of the supply chain and delivery systems [18]. Although the successful assembly of modules depends on other upstream stages in the VMC supply chain, the OA stage is often inherited with symbiotic risk events and processes. The assembly of modules requires the effective lift path planning of the cranes and layout planning of the site (OA-1); however, previous studies reported incidents where OA-1 has significantly hindered the success of the VMC project [91]. The crane selection and operation are essential tasks to manage optimization processes, analysis of the lifting cycle and probable collision; thus, lift planning, predominantly an iterative trial and error mechanism, becomes crucial for assembling modules. Improper lift planning of the cranes can result in poor stability, blind lifting, and the breakdown of cranes (OA-2) [92]. Figure 9 shows the PARETO chart for onsite assembly risks. The five CRFs, namely OA-1, OA-2, OA-4, OA-3, and OA-7, are vital and responsible for 80% of the effect on the onsite assembly stage. Thus, these CRFs entail additional attentiveness during the onsite assembly of the VMC modules.

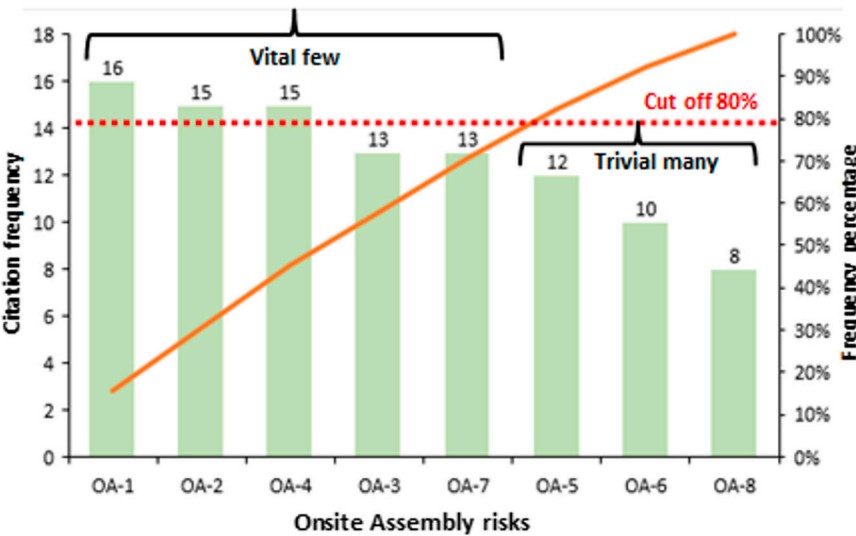

**Figure 9.** PARETO chart for onsite assembly risks in VMC.

Moreover, the proper analysis and planning for module specification information in terms of weight, height and volume are necessary to avoid extra load on the crane, further leading to cables breaking or jig falling (OA-3) [99]. Additionally, the correct optimization of other site activities requires effective layout planning to enhance productivity at the site and minimize the safety risks likely to occur (OA-1). The lift and layout planning

incompetency are reported in dense urban environments, where the margin of error is minimal and incurred losses can be superfluous [99].

Other risk events caused by unsafe acts and conditions by the workers onsite have also been cited in previous research, and show the workers' inabilities to deal with successful VMC projects (OA-4) [25]. For instance, in 2019, the Bureau of Labor Statistics reflected that the rate of injuries and incidents was 10.2 in 100 workers in VMC, compared to 5.2 in 100 in traditional construction methods [8]. Moreover, in their study, Fard et al. [25] reported 125 cases of accidents caused by the VMC construction method, of which 38.4% were fatal. A few unsafe acts on the assembly site are reported due to manual lifting, unwrapping, lining, unhooking, screwing, and welding (OA-6) [12].

In terms of unsafe or untimely conditions, the effects of wind and weather and near environmental disruptions were also reported as a severe risk during the OA stage (OA-8) [18]. Although many risks at the assembly stage occur due to site uncertainties, a few others result from actions taken in the upstream stages of VMC. One such issue is the variation in geometry and dimensions of the module from the production stage (OA-7) [71]. This discrepancy is due to the neglect of the design process. It is further exacerbated within the factory setting, where comparatively low-experience fabricators produce modules with errors in dimensions and geometry [16]. Due to that, the alignment of the modules is poor when assembled onsite, resulting in a shortcoming in precision and accuracy. This affects the module' horizontal and vertical alignment and compromises site fit requirements. Another issue is the poor verification of the modules due to inadequate tagging from the production stage (OA-5) [22]. Upon receiving the modules, it is necessary to check the modules based on working or shop drawings provided by the design team; therefore, proper tagging of the modules becomes essential, as manual checking consumes more time, going against tight schedules [18].

In most cases, the modules are attached with radio frequency identification (RFID) and global positioning system (GPS) tags, which the site operator can retrieve information from upon receiving the module [93]. However, it has been reported that a single source of auto-tags is ineffective in retrieving the real-time information of the module, which carries the BIM geometrical and metadata for the module. Further, these tags can sometimes get broken or damaged due to transportation glitches, making it even more challenging to retrieve the correct information [17]. In this regard, JIT delivery is suitable, along with smart trinity tags that include multiple sources of tags, so if one gets broken or damaged, there is always another to deliver the communication [48]. However, due to the high cost incurred, smart trinity tags are not often used, and they require smart receiver tags on the site to be effective.

Further, a low onsite buffer and hedging space cause other risks, such as misplacement of modules, obscurity in identifying the modules, and damage to the modules [17]. The flawed process of welding in the modules is also an issue predominantly caused by the manufacturer' lack of capability during manufacturing modules. Nevertheless, the OA stage is crucial for the success of the VMC project, and timely measures should be applied to reduce them [18]. The use of virtual simulation tools to train the workers onsite, robotic mechanisms for cumbersome tasks, and pre-optimization of the crane lift planning and site layout planning should be implemented to manage the risks at the onsite assembly stage.

### 4.3. Project Attribute Risks

#### 4.3.1. Implementation and Schedule

The disruptive nature of VMC fetches new tangents of uncertainties in the construction process. This brings unique challenges to the decision-makers and stakeholders involved. At a basic level, VMC is awaiting broad adoption and still has resistance in many regions due to an assortment of factors [52]. The implementation risks are caused due to various reasons depending on the region's geographical location, socio-economic factors, and socio-political issues. In Mainland China and Hong Kong, although VMC is fledgling yet escalating, owing to their new policies and motivation due to the growing population,

the risks pertaining to implementation are manifold, and adoption at a broader scale is meagre [51]. Luo et al. [44] mentioned an inefficient interface between multiple parties, lack of design codes and standards, the high capital cost involved, lack of modern technologies for the monitoring process, and inadequate management techniques as the five reasons causing implementation problems of the VMC. Further, the lack of novel business models for VMC, the complex nature of the events involved and insufficient knowledge about the process are risk factors owing to the low implementation and adoption of VMC at a significant scale. Overall, all these risks render the low adoption of VMC and become the recipe for low interest from the stakeholders, thus causing implementation problems.

Moreover, the schedule is one of the significant steps in any project, and delays in the schedule occur when the stipulated time of the project exceeds the time frame decided. Although delays are sometimes unavoidable in a construction project, the risks in VMC are more likely due to its multifarious nature [26]. Like other types of risks, schedule risks are diverse and can occur at any project stage. The disruptions in the supply chain process, inefficient worker' experience, limited best practice guidelines, and inadequate component connections of the modules are some reasons for schedule delays [58].

Further, slow quality inspection procedures, inefficient design data transition, and logistics information inconsistency delay the process at the manufacturing stage [16]. Besides these risks, some unwanted factors such as weather disruptions, wind interruptions, and natural phenomena can also result in schedule delays, thus affecting the productivity of the VMC process [21]. Therefore, these risks should be carefully considered before the commencement of the project to avoid delays and, eventually, the project's productivity.

### 4.3.2. Supply Chain and Financial

The fragmented nature of the VMC supply chain is due to the complex nature of the stages involved. The synchronization of the various stages, such as design, manufacturing, transportation, storage, and installation, needs to be implemented precisely and effectively [40]. The decisions at each project stage involve several uncertainties, requiring a collaborative approach. Initially, during the design phase, it is imperative to have a clear justification and plan to use VMC. The involvement of different participants is significant, as the decisions or changes made at this stage are obscure for the downstream stages [61]. The choice of a modular manufacturer, factory location, accessibility of the land, procurement of materials, and means of transportation should be finalized appropriately before the start of the manufacturing process [16].

Moreover, as the VMC process is usually based on an engineer-to-order operation, the bidding decisions should be finalized the earliest to avoid problems related to site inventory. The interdependencies of the events require novel configurations to avoid any hindrances in the supply chain, as the disturbance can occur at any project stage [54]. Risks such as module fabrication failure, inaccuracies in the components, and onsite equipment failure may halt the process, especially as these risks cannot be predicted until they happen [46]. The management process requires unique methods and a collaborative approach, as the stakeholders in the VMC process have challenges and tasks to perform. In this regard, BIM automation of the process and visualization techniques must be made and supported [108]. The whole supply chain process can be modeled in a game environment, from design to manufacturing to transportation to installation [105]. Each stakeholder should get training on the possible risks that might occur at various stages.

Regarding financial risks, adopting the VMC method requires enormous capital to provide the proper production and supply of modules and the related components. The cost involves buying space for the offsite factory, the equipment for production purposes, the manufacturing plant, and materials for modules and workers [109]. The high capital investment in the VMC process makes investors reluctant to it, as the risks involved in the VMC process are still new and can cause considerable losses in terms of the money involved [110]. Different countries have several reasons depending on socio-economic and socio-political factors, making it challenging to adopt the VMC process. For instance,

Li et al. [42] listed the volatile climate of Canada as a reason for the unwillingness of stakeholders to adopt the VMC approach.

Similarly, Lee and Kim [76] report factors such as inadequate expertise in the design of modular components, inefficient cost estimation, unsteady rate of module production, and inaccuracies in the design of the structure for the escalation of capital within the South Korean market, which instigates reluctance to adopt VMC approaches. On the other hand, in China, the lack of codes and modern techniques, hesitation of public consumption, and escalated module prices limit the investment process [23]. Oceanic countries such as Australia and New Zealand lack the adoption of VMC methods due to the module' transportation costs due to the cross-border supply chain [40]. Therefore, investment risks are bound and differ from country to country.

### 4.3.3. Safety and Ergonomics

The construction process is usually considered risky, exposing workers and other related staff to potential health and safety threats. Issues such as unsafe acts and unsafe conditions give rise to several risks, such as falling, awkward working positions, and work-related musco-skeletal disorders (WMSDs) [25,76]. Despite numerous VMC advantages, the Bureau of Labor Statistics in the USA reported injuries and incidence rates during VMC projects to be 10.2 persons per 100, which is higher than 5.2 persons per 100 in CCMs [111]. This is due to the unique processes of VMC, which tends to produce uninformed risks. Workers are prone to risks at different stages of the VMC process. For instance, exposure to harmful mineral fibers, combustion fumes, furnace heat, asbestos, and arsenic at factory sites can result in lung-related problems [25].

Additionally, problems such as exposure to noise, sawdust, organic compounds, and other harmful raw material, become the recipes for worker' health and safety hazards [76]. Other than this, awkward postures of the workers during the module fabrication process may result in spine injuries and low back pain injuries [75]. Previous studies have applied different techniques to automate the process offsite and onsite, but a wide acceptance of those studies on real projects is still missing [112]. Techniques such as robotic automation in factories integrated with the internet of things (IoT) have been implemented to reduce health and safety risks [106]. Darko et al. [30] suggested the significance of immersive techniques in getting knowledge and training about these health and safety risks through game-based learning. As the workers involved in the VMC process are still lacking a high experience level of this process, the use of BIM-based game learning could be the strategy to equip the workers and related staff to get hands-on training and assessment of their skills before getting involved in the task prone to risks related to health and safety.

### 4.3.4. Civil and Structural

Continuous changing climate results in alteration in the geology of the earth's surface, which requires different structural obligations each time, prodding the risks associated with it [113]. The complexity in the structure for VMC projects is significantly becoming challenging, especially in areas with congested neighborhoods such as Singapore and Hong Kong. Although the VMC process needs less time than conventional construction, a final product can result in poor and inefficient structural integrity [26].

In addition, the eccentricities in floor slabs and other structure members could complicate the process resulting in schedule delays, cost overruns and diminished quality. In addition, some modular buildings made of timber exhibited the problem of poor thermal insulation and overheating of the buildings [86]. Problems such as mold growth, dampness, condensation, and improper insulation have also been cited by previous researchers in modular buildings [46]. Existing methods to resolve this risk depend on trial-and-error solutions and other similar time-taking and inefficient methods. Any discrepancies in the structural integrity and solidity should be assessed and managed before commencement of the work.

## 5. Mitigation Framework for Using Digital-Technology-Based Circular Strategies to Overcome VMC Risks

Digital technologies (DTs) and circular economy (CE) are the two most significant industrial patterns that have epitomized the Industrial Revolution 4.0 (IR-4.0) in the recent years [114]. The amalgamation of these two delivers the context of improving efficiency, effectiveness, and success in the corresponding relative field, including construction.

Since the dawn of the Industrial Revolution 4.0 (IR-4.0), the growth of DTs and CE has escalated in various sectors, including construction [114]. However, the construction industry is still slow in adopting DTs and CE, especially in the VMC method. Nevertheless, implementing DTs is indispensable for the overall automation of the VMC process at all stages. Further, it has been reported in a recent study that the automation of VMC through DTs will not only enhance its productivity, but also strengthen sustainability and circularity throughout the process [112]. This section will highlight the application of DTs to solve VMC risks and reflect some of the CE principles that can be exploited and integrated with the VMC process at different stages. These strategies were extracted from the extant literature that has implicitly and tacitly mentioned them. Moreover, Figure 10 illustrates the mitigation strategies for each project stage and project attribute risks, highlighting the relevant DT for each category.

Initially, at the DP stage, BIM as a digital technology reduces the inefficiencies in information sharing, thus minimizing data wastage and aiding in optimizing the VMC building design, further reducing resource and waste creation [61]. Moreover, as a digital model, BIM facilitates end-of-life (EoL) design and develops a project database for circularity assessment and material information. This material information can be used for the recovery process, and maintaining digital material passports (DMPs) and data banks for the VMC project [114]. Additionally, BIM plugins or add-ins are used to predict reusability, recyclability, disassembly process, and end-of-life performance of a VMC building, estimating the design waste, and thus reducing them at the earliest stages [115]. Additionally, using parametric design coupled with machine learning (ML) algorithms optimizing design alternatives can create an efficient BIM model for the DfMA-based VMC building through early design considerations for slowing and closing the resource loops as a CE strategy [116].

Along with several design strategies, design for disassembly is crucial for achieving CE in the VMC project; thus the BIM model should incorporate design for disassembly guidelines [117]. Design for disassembly can be facilitated by utilizing DfMA principles within the BIM model, as the fundamental CE strategy is to recycle and reuse the material at the end-of-life (EoL) [117]. Moreover, coupled with artificial intelligence (AI), Internet of Things (IoT) devices, and using Big Data Analytics (BDA), the BIM model can predict the carbon footprint, resource exploitation, potential defects, and performance criteria for the VMC building [115]. BDA deployment in the early stages of design also facilitates low-carbon VMC design and predicts energy performance, thus reducing the operational costs during the downstream stages of VMC [118].

Additionally, block chain technology (BCT) based on secured information storage for material passports in data banks and digital platforms provides ground for tracking the digital twin of a BIM model with geometric and non-geometric data, thus closing the data wastage loop [114]. Another strategy in the DP stage is the use of bio-based materials that can be used to develop module' prototypes using additive manufacturing (AM), thus reducing safety issues and accuracy concerns, and helping optimize modules during the early stages [119].

| | Design and Planning | Offsite Manufacturing | Transport and Logistics | Onsite Assembly |
|---|---|---|---|---|
| **Strategies for reducing and minimising MiC project stage risks** | ❖ BIM reduce data wastage, optimise design, minimise resource and waste<br>❖ BIM facilitates EoL and maintaining of DMP for circularity assessment and syncing of the DT<br>❖ AI and ML-based parametric design for optimising and getting the best design and planning options<br><br>❖ BIM aids DfMA practice to facilitate lifecycle loops for circularity<br>❖ BDA-IoT-based carbon footprint optimization of modules design<br>❖ BCT for managing data banks and DMPs to reduce initial data waste<br><br>❖ AM for developing prototypes of modules with bio-based material | ❖ BIM-based DfMA shop drawings of modules make them customisable for circular use<br>❖ RB and Dfab increase recycled and reusable materials use, reduce injury and deliver safe tasks<br>❖ AM and 3D printing utilises renewable materials to develop modules<br>❖ IoT and RFID facilitate material knowledge used in the EoL management<br><br>❖ BCT-based DMP makes reverse logistics and recycling efficient<br>❖ AGVs use makes the factory environment safe and reduces injuries<br>❖ BCT facilitates smart data register | ❖ BIM for structural health monitoring of modules during transport facilitate fast response<br>❖ IoT-BDA data-driven monitor to detect possible damages reducing transport disruptions<br>❖ BCT real-time traceability of the modules facilitating the quick approval process<br>❖ AI-based transport cost estimation to increase the effectiveness and reduce inefficiency in module clearance<br>❖ IoT-based global damage detection for modules during transportation<br><br>❖ AGVs use to reduce worker safety risks and quick transportation | ❖ VR and AR for pre-optimisation of site layout planning, reducing space wastage<br>❖ Integrative approach of lean construction and Virtual Design and Construction reduces construction site wastage<br>❖ BIM-AI integrative approaches for safety, and schedule management, thus processing complex data<br>❖ AM and 3D printing-based formwork for modules further utilises recycled and reusable materials on site<br>❖ BIM-IoT-based material passport for increasing EoL of virgin materials<br>❖ BCT enabled cyber-physical smart construction for data to utilise |
| **Enabling digital technologies** | BIM – AI – IoT – BDA – BCT – DT – AM - ML | BIM – RB – AM – AGV – IoT – BCT – DMP - RFID | BIM – AI – IoT – BDA – BCT – ML - AGV | BIM – AM – RB – BCT – IoT – AI – VR - AR |
| **Project stage** | Design and Planning | Offsite Manufacturing | Transport and Logistics | Onsite Assembly |
| **Project attribute** | Implementation and Schedule | Supply chain and Financial | Safety and Ergonomic | Civil and Structural |
| **Enabling digital technologies** | BIM – RFID – GPS – AR – VR – AI – ML – BDA - IoT | BIM – RFID – GPS – IoT – AI – BDA – BCT – DMP - ML | BIM – RFID – IoT – AR – VR – AI – AM – ML - BDA | BIM – IoT – RFID – AR – AI – ML – BDA - DT |
| **Strategies for reducing and minimising MiC project attribute risks** | ❖ BIM for quality detection of the modules with real-time data<br>❖ RFID tracking for the material and objects used in the factory and site<br>❖ GPS-based information collection of modules compared to the BIM model<br>❖ AR-based inspection of modules for reducing errors and repair the tasks<br>❖ VR use for the factory production process and onsite real-time construction progress<br>❖ AI and ML models for predicting and optimising construction tasks<br><br>❖ IoT-BDA for collecting, optimising, and realisation of item-level data for the developed modules | ❖ BIM visualisation and streamlining of logistics, module production, and onsite assembly<br>❖ RFID-based material tracking status and identification in near real-time<br>❖ GPS-based transportation route assessment, module loading and unloading data in BIM model environment<br>❖ IoT-based traceability and visibility of material, module, and objects<br>❖ AI for supply chain optimisation and using ML models to predict best-fit production & onsite process<br>❖ BCT-based DMP library & register for data tracking for EoL use<br>❖ BDA for modules data for lifecycle assessment | ❖ BIM-based 4D simulation for tasks in the factory and onsite assembly stage<br>❖ IoT-integrated RFID tags for smart construction objects installed on workers – helmets, armbands etc.<br>❖ AR-assisted risk identification and modelling for production tasks<br><br>❖ VR-based crane simulation, lift and layout planning for onsite tasks<br><br>❖ AI-assisted precautions and early notification of dangerous tasks and hazards in the factory and onsite<br>❖ AM and 3DP for arduous and strenuous tasks to minimise the injury risks for workers<br>❖ BDA-ML to predict hazards on site | ❖ BIM-based modelling for reducing dimensional and geometrical inaccuracies and tolerances<br>❖ IoT and RFID for structural health monitoring of modules during the transport process<br>❖ AR-based assistance for looking at and optimising module eccentricities<br>❖ AI-based ML models for predicting the deterioration of modules during the transportation process to onsite<br><br>❖ BDA and IoT data-driven monitoring system to detect damage and vibration analysis of the modules<br>❖ BIM and AI-based development of digital twin for modules SHM |

NOTE: BIM – Building Information Modelling, AI- Artificial Intelligence, IoT – Internet of Things, BDA – Big Data Analytics, BCT – Block Chain Technology, DT – Digital Twin, AM – Additive Manufacturing, ML – Machine Learning, RB – Robotics, AGV – Automated Guided Vehicles, DMP – Digital Material Passport, RFID – Radio Frequency Identification, VR – Virtual Reality, AR – Augmented Reality, and; GPS – Global Positioning System

**Figure 10.** Risk mitigation framework for the utilization of digital-technology-based circular strategies for VMC risks.

In factory production, digital fabrication and robotics are energy-saving methods for building module components, thus increasing the use of recycled and reusable materials, reducing worker injuries, and providing safer environments [114]. Furthermore, AM/3D printing can be utilized to develop modules from recycled concrete and other bio-based materials to minimize the waste of resources from human labor [119]. Further, AM/3D printing facilitates the consumption of resins and substrates in the modules developed from reused and renewable materials [120].

Moreover, installing sensing and IoT devices in the module components during manufacturing facilitates the knowledge of material quality that can be further deployed at the EoL [121]. Similarly, a BCT-based smart product data register could be maintained on digital platforms, aiding the resource utilization and allocation process during production [122].

VMC modules are predominantly made of concrete and steel: both can be recycled and reused in producing new modules using the AM process [119]. On the one hand, steel can be recycled multiple times without losing mechanical properties, and concrete can be crushed into inert material to be used during production [112]. A data repository of used materials could be maintained on the BCT network to allow them to be further leased and utilized after EoL [114]. The BCT-based smart contract should be maintained, specifying reverse logistics delivery options to recycle and reuse materials [115].

Furthermore, the biggest threat at the TL stage is the module' structure health monitoring (SHM), which has resulted in severe delays and disruption of VMC projects [123]. Digital technologies can be utilized for effective SHM of the modules for defect-free delivery and quick response to any failure or faults in the prefabricated modules. The vibration data during the transportation of the modules can be utilized by installing IoT-based global damage detection sensors, thus making it more cost-effective and time efficient to realize the possible defects in the tolerance and fit-outs of the modules [123].

In addition, AI-based transportation cost estimation can be significant to learn from past events, such as port stoppages, as in the case of cross-country transportation of modules, and inefficiency caused during the customs clearance [17]. Better-informed decisions can be facilitated using AI-based techniques for distance determination between a construction site and factory location to reduce operational costs [123]. In addition, IoT-based data-driven monitoring systems detected possible damages that occurred due to the transport of overweight modules and technical problems with the vehicles, thus reducing transport disruptions [124].

Regarding the utilization of BCT during the TL stage, the BCT-based supply chain management platform analyzes the module' information, and can thus reduce costs with better clarity and traceability [125]. Module information sharing and near real-time traceability can be realized using a BCT-based smart contract addressing the security issues and having a quick approval process [121]. Finally, BDA is integrated with a sensor-based data acquisition model system to record and store the acceleration data produced by the modules during the transportation process, aiding decision-making and reducing data loss [123].

During the OA stage, digital technologies such as virtual and augmented reality can be used for the pre-optimization of site layout planning in high-rise VMC projects in dense urban environments, to reduce the wastage of space [92,99]. Moreover, the amalgamation of lean construction and virtual design and construction (VDC) techniques effectively reduces construction wastage at the site [48]. Further, safety management, quality management, and schedule risk identification can be facilitated and optimized by utilizing AI techniques with BIM, thus processing complex data and minimizing the loss of information [126].

Additionally, AM and 3D-printing-based formwork can be made onsite using recycled and reusable materials, thus avoiding site waste and environmental pollution [119]. Likewise, BIM-based material passports should be allocated to used components, and the information should be uploaded to relevant databanks, thus making their utilization at the end of life [114]. Alongside that, IoT-based smart construction objects (SCOs) can be

deployed to track materials, labor, and equipment to enhance scheduling with limited time and cost wastage [124].

The management of different assets at the site can be facilitated by developing a digital twin of the onsite assembly process that can better detect logistics for the elements [127]. Likewise, human–robot interaction (HRC)-based semi-automation process should be deployed to reduce the onsite safety risks, and further enhance work productivity and save time [128]. Finally, the OA stage can be befitted by the BCT-enabled smart cyber-physical platform that can be developed onsite to allow better information sharing and integration among stakeholders, thus minimizing data fragmentation and discontinuity in the VMC process [122].

## 6. Conclusions

The COVID-19 pandemic has further highlighted the needs for the construction industry to find innovative solutions to the long-lasting problem of productivity in the sector. Although existing for quite some time, the rise in VMC has fluctuated due to its unique design principles, engineering procedures, supply chain activities, web of stakeholder composition, and other management requirements. Responding to this issue, many theoretical and empirical studies have examined VMC risks and vulnerabilities; however, a holistic understanding of risks in terms of project stage and project attributes is missing and required. Therefore, this study identified the risks in VMC through the lens of project stage and project attribute risks. Through a mixed review process, including systematic literature review and critical content analysis, this study ponders over 91 peer-reviewed journal articles to explore the current research relating VMC risks.

The systematic literature analysis revealed the publication trend of VMC risks that have escalated over the recent years, relevant journals that have contributed, geospatial distribution of the studies, and important keywords for the selected studies. Further, the content analysis generated two categorical classifications of VMC risks, namely project stage and project attribute. Thirty-one CRFs for project stage risks within DP, OM, TL, and OA stages were listed and analyzed. DP stage risks were found to be significant in the overall supply chain of the VMC project, with the most effect on the upstream and downstream stages. Further, project attribute risks were classified and thoroughly discussed within the categories of implementation and schedule, supply chain and financial, safety and ergonomic, and civil and structural risks.

The study has both theoretical and practical implications. Theoretically, the study contributes to the body of knowledge by delivering a checklist of critical risk factors within the VMC stages, and guides towards understanding those risks in detail. Practically, the study delivers the potential of digital-technology-based circular strategies to overcome the VMC risks that can be useful for various stakeholders of the VMC industry.

Although the study successfully realized the objectives framed, the following limitations are worth highlighting for future refinements. Firstly, the study utilized the frequency of citations to determine the risk criticalities and was hampered by the lack of detailed empirical data that may not reflect the context significance. Further, the synthesis of digital-technology-based circular mitigation strategies were retrieved from the reviewed studies which tacitly and implicitly mentioned them. Intrinsically, a quantitative assessment and validation using expert opinion will be helpful for a more comprehensive understanding of the risks and mitigation strategies associated with VMC in future studies. Secondly, the sweeping generality of the risk factors is against the geographical sensitivity that may differ based on policies, strategies, and industry standards. Nonetheless, along with providing bespoke explanations of different CRFs in each stage and a few measures to overcome them, the study has progressed the debate towards developing better strategies to overcome VMC risks in light of digital technologies coupled with circular economic principles, aiming to achieve the United Nations sustainable development goals by 2030 toward a more energy-efficient and sustainable planet.

**Author Contributions:** Conceptualization, A.A.K.; methodology, A.A.K.; software, A.A.K.; validation, A.A.K.; formal analysis, A.A.K.; investigation, A.A.K.; resources, A.A.K.; data curation, A.A.K.; writing—original draft preparation, A.A.K.; writing—review and editing, A.A.K., R.Y. and T.L. visualization, A.A.K.; supervision, R.Y., T.L., N.G. and J.W.; project administration, R.Y.; funding acquisition, R.Y. All authors have read and agreed to the published version of the manuscript.

**Funding:** This research received no external funding.

**Institutional Review Board Statement:** Not applicable.

**Informed Consent Statement:** Not applicable.

**Data Availability Statement:** All the data used is mentioned.

**Acknowledgments:** The study acknowledges the University of South Australia for providing a PhD scholarship for the first and corresponding author.

**Conflicts of Interest:** The authors declare that they have no known competing financial interests or personal relationships that could have appeared to influence the work reported in this paper.

## Appendix A

**Table A1.** List of 91 articles used in the current study.

| Serial Number | Paper Title | Year | Source Title |
|---|---|---|---|
| 1 | A Proactive Risk Assessment Framework to Maximize Schedule Benefits of Modularization in Construction Projects | 2023 | *Journal of Construction Engineering and Management* |
| 2 | VR—MOCAP-Enabled Ergonomic Risk Assessment of Workstation Prototypes in Offsite Construction | 2023 | *Journal of Construction Engineering and Management,* |
| 3 | Assessing the Off-Site Manufacturing Worker' Influence on Safety Performance: A Bayesian Network Approach | 2023 | *Journal of Construction Engineering and Management,* |
| 4 | Safety Risk Management of Prefabricated Building Construction Based on Ontology Technology in the BIM Environment | 2022 | *Buildings* |
| 5 | Digital Twin-Based Intelligent Safety Risks Prediction of Prefabricated Construction Hoisting | 2022 | *Sustainability* |
| 6 | Analysis on risk factors related delay in PCPs | 2022 | *Engineering, Construction and Architectural Management* |
| 7 | Understanding Disputes in Modular Construction Projects: Key Common Causes and Their Associations | 2022 | *Journal of Construction Engineering and Management* |
| 8 | Use of Virtual Reality to Assess the Ergonomic Risk of Industrialized Construction Tasks | 2022 | *Journal of Construction Engineering and Management,* |
| 9 | A critical analysis of benefits and challenges of implementing modular integrated construction | 2022 | *International Journal of Construction Management* |
| 10 | Understanding the Key Risks Affecting Cost and Schedule Performance of Modular Construction Projects | 2022 | *Journal of Management in Engineering* |
| 11 | Research on the rework risk core tasks in prefabricated construction in China | 2022 | *Engineering, Construction and Architectural Management* |
| 12 | Identification of critical factors influencing prefabricated construction quality and their mutual relationship | 2022 | *Sustainability* |
| 13 | Managing stakeholder-associated risks and their interactions in the life cycle of prefabricated building projects: A social network analysis approach | 2022 | *Journal of Cleaner Production* |

**Table A1.** *Cont.*

| Serial Number | Paper Title | Year | Source Title |
|---|---|---|---|
| 14 | Barriers to the development of prefabricated buildings in China: a news coverage analysis | 2022 | *Engineering, Construction and Architectural Management* |
| 15 | Overcoming process-related barriers in modular high-rise building projects | 2022 | *International Journal of Construction Management* |
| 16 | Critical factors for successful implementation of just-in-time concept in modular integrated construction: A systematic review and meta-analysis | 2022 | *Journal of Cleaner Production* |
| 17 | Critical considerations on tower crane layout planning for high-rise modular integrated construction | 2022 | *Engineering, Construction and Architectural Management* |
| 18 | Integrating critical chain project management with last planner system for linear scheduling of modular construction | 2022 | *Construction Innovation* |
| 19 | A quantitative assessment of greenhouse gas (GHG) emissions from conventional and modular construction: A case of developing country | 2022 | *Journal of Cleaner Production* |
| 20 | Sources of Uncertainties in Offsite Logistics of Modular Construction for High-Rise Building Projects | 2022 | *Journal of Management in Engineering* |
| 21 | The influence of government's economic management strategies on the prefabricated buildings promoting policies: Analysis of quadripartite evolutionary game | 2022 | *Buildings* |
| 22 | Empirical Study of Identifying Logistical Problems in Prefabricated Interior Wall Panel Construction | 2022 | *Journal of Management in Engineering* |
| 23 | Exploring the critical production risk factors for modular integrated construction projects | 2022 | *Journal of Facilities Management* |
| 24 | Multi-criteria decision analysis for tower crane layout planning in high-rise modular integrated construction | 2022 | *Automation in Construction* |
| 25 | Heavy mobile crane lift path planning in congested modular industrial plants using a robotics approach | 2022 | *Automation in Construction* |
| 26 | Critical supply chain vulnerabilities affecting supply chain resilience of industrialized construction in Hong Kong | 2022 | *Engineering, Construction and Architectural Management* |
| 27 | Computer vision-based disruption management for prefabricated building construction schedule | 2022 | *Journal of Computing in Civil Engineering* |
| 28 | Analysis of safety risk factors of modular construction to identify accident trends | 2022 | *Journal of Asian Architecture and Building Engineering* |
| 29 | Digital twin for supply chain coordination in modular construction | 2022 | *Applied Sciences* |
| 30 | Risk-Based Approach to Predict the Cost Performance of Modularization in Construction Projects | 2022 | *Journal of Construction Engineering and Management* |
| 31 | Predicting delays in prefabricated projects: SD-BP neural network to define effects of risk disruption | 2022 | *Engineering, Construction and Architectural Management* |
| 32 | Research on investment risk influence factors of prefabricated building projects | 2021 | *Journal of Civil Engineering and Management* |
| 33 | Worker' safety behaviors in the off-site manufacturing plant | 2021 | *Engineering, Construction and Architectural Management* |

**Table A1.** *Cont.*

| Serial Number | Paper Title | Year | Source Title |
|---|---|---|---|
| 34 | Constraints hindering the development of high-rise modular buildings | 2021 | *Applied Sciences* |
| 35 | Comparative analysis of modular construction practices in mainland China, Hong Kong and Singapore | 2021 | *Journal of Cleaner Production* |
| 36 | Comparison of Worker Safety Risks between Onsite and Offsite Construction Methods: A Site Management Perspective | 2021 | *Journal of Construction Engineering and Management* |
| 37 | Exploring the status, benefits, barriers and opportunities of using BIM for advancing prefabrication practice | 2021 | *International Journal of Construction Management* |
| 38 | Barriers to the adoption of modular integrated construction: Systematic review and meta-analysis, integrated conceptual framework, and strategies | 2021 | *Journal of Cleaner Production* |
| 39 | Critical factors influencing the sustainable construction capability in prefabrication of Chinese construction enterprises | 2021 | *Sustainability* |
| 40 | Modelling the critical risk factors for modular integrated construction projects | 2021 | *International Journal of Construction Management* |
| 41 | Stochastic-based noise exposure assessment in modular and off-site construction | 2021 | *Journal of Cleaner Production* |
| 42 | Building information modeling (BIM)-based modular integrated construction risk management—Critical survey and future needs | 2021 | *Computers in Industry* |
| 43 | Dynamic and Proactive Risk-Based Methodology for Managing Excessive Geometric Variability Issues in Modular Construction Projects Using Bayesian Theory | 2021 | *Journal of Construction Engineering and Management* |
| 44 | Multi-agent simulation for managing design changes in prefabricated construction projects | 2021 | *Engineering, Construction and Architectural Management* |
| 45 | Identifying supply chain vulnerabilities in industrialized construction: an overview | 2021 | *International Journal of Construction Management* |
| 46 | Game analysis on prefabricated building evolution based on dynamic revenue risks in China | 2021 | *Journal of Cleaner Production* |
| 47 | Risks of modular integrated construction: A review and future research directions | 2021 | *Frontiers of Engineering Management* |
| 48 | Risks in Prefabricated Buildings in China: Importance-Performance Analysis Approach | 2021 | *Sustainability* |
| 49 | Factors influencing the application of prefabricated construction in China: From perspectives of technology promotion and cleaner production | 2020 | *Journal of Cleaner Production* |
| 50 | Monte Carlo simulation for tolerance analysis in prefabrication and offsite construction | 2020 | *Automation in Construction* |
| 51 | Barriers to Building Information Modeling (BIM) implementation in China's prefabricated construction: An interpretive structural modeling (ISM) approach | 2020 | *Journal of Cleaner Production* |
| 52 | Critical risk factors in the application of modular integrated construction: a systematic review | 2020 | *International Journal of Construction Management* |

**Table A1.** *Cont.*

| Serial Number | Paper Title | Year | Source Title |
|---|---|---|---|
| 53 | Stakeholder-Associated Supply Chain Risks and Their Interactions in a Prefabricated Building Project in Hong Kong | 2020 | *Journal of Management in Engineering* |
| 54 | Risk-averse supply chain for modular construction projects | 2020 | *Automation in Construction* |
| 55 | Opportunities and challenges of modular methods in dense urban environment | 2020 | *International Journal of Construction Management* |
| 56 | Managing information flow and design processes to reduce design risks in offsite construction projects | 2020 | *Engineering, Construction and Architectural Management* |
| 57 | Integrated Risk Management Framework for Tolerance-Based Mitigation Strategy Decision Support in Modular Construction Projects | 2020 | *Journal of Management in Engineering* |
| 58 | Exploring the interactions among factors impeding the diffusion of prefabricated building technologies: Fuzzy cognitive maps | 2020 | *Engineering, Construction and Architectural Management* |
| 59 | Perceptions towards risks involved in off-site construction in the integrated design & construction project delivery | 2020 | *Journal of Cleaner Production* |
| 60 | Assessing and Prioritizing Delay Factors of Prefabricated Concrete Building Projects in China | 2020 | *Applied Sciences* |
| 61 | Barriers to promoting prefabricated construction in China: A cost–benefit analysis | 2019 | *Journal of Cleaner Production* |
| 62 | A model for simulating schedule risks in prefabrication housing production: A case study of six-day cycle assembly activities in Hong Kong | 2019 | *Journal of Cleaner Production* |
| 63 | The hindrance to using prefabrication in Hong Kong's building industry | 2019 | *Journal of Cleaner Production* |
| 64 | Key constraints and mitigation strategies for prefabricated prefinished volumetric construction | 2019 | *Journal of Cleaner Production* |
| 65 | Applying internal insulation in post-war prefab housing: Understanding and mitigating the hygrothermal risks | 2019 | *Building and Environment* |
| 66 | Barriers to the transition towards Off-site construction in China: An Interpretive Structural Modeling approach | 2019 | *Journal of Cleaner Production* |
| 67 | Constraints on the promotion of prefabricated construction in China | 2019 | *Sustainability* |
| 68 | Overcoming barriers to off-site construction through engaging stakeholders: A two-mode social network analysis | 2019 | *Journal of Cleaner Production* |
| 69 | Identifying barriers to off-site construction using grey DEMATEL approach: Case of China | 2019 | *Journal of Civil Engineering and Management* |
| 70 | Schedule delay analysis of prefabricated housing production: A hybrid dynamic approach | 2018 | *Journal of Cleaner Production* |
| 71 | Research on investment risk management of Chinese prefabricated construction projects based on a system dynamics model | 2018 | *Buildings* |
| 72 | Integrating RFID and BIM technologies for mitigating risks and improving schedule performance of prefabricated house construction | 2018 | *Journal of Cleaner Production* |

**Table A1.** *Cont.*

| Serial Number | Paper Title | Year | Source Title |
|---|---|---|---|
| 73 | Critical factors affecting the quality of industrialized building system projects in China | 2018 | *Sustainability* |
| 74 | Schedule risk modeling in prefabrication housing production | 2018 | *Journal of Cleaner Production* |
| 75 | Safety concerns related to modular/prefabricated building construction | 2017 | *International Journal of Injury Control and Safety Promotion* |
| 76 | Managing risk in modular construction using dimensional and geometric tolerance strategies | 2017 | *Automation in Construction* |
| 77 | Analysis of cost increasing risk factors in modular construction in Korea using FMEA | 2016 | *KSCE Journal of Civil Engineering* |
| 78 | Thermal comfort, summertime temperatures and overheating in prefabricated timber housing | 2016 | *Building and Environment* |
| 79 | Schedule risks in prefabrication housing production in Hong Kong: a social network analysis | 2015 | *Journal of Cleaner Production* |
| 80 | Major Barriers to Off-Site Construction: The Developer's Perspective in China | 2015 | *Journal of Management in Engineering* |
| 81 | Risk factors affecting practitioners' attitudes toward the implementation of an industrialized building system: A case study from China | 2015 | *Engineering Construction and Architectural Management* |
| 82 | Risk assessment and management practices (RAMP) within the Tanzania construction industry: Implementation barriers and advocated solutions | 2014 | *International Journal of Construction Management* |
| 83 | Barriers of Implementing Modern Methods of Construction | 2014 | *Journal of Management in Engineering* |
| 84 | Factors impeding the offsite production of housing construction in China: An investigation of current practice | 2014 | *Construction Management and Economics* |
| 85 | Exploring the challenges to industrialized residential building in China | 2014 | *Habitat International* |
| 86 | An Investigation of Critical Factors and Constraints for Selecting Modular Construction Over Conventional Stick-Built Technique | 2013 | *International Journal of Construction Education and Research* |
| 87 | Risk identification and assessment of modular construction utilizing fuzzy analytic hierarchy process (AHP) and simulation | 2013 | *Canadian Journal of Civil Engineering* |
| 88 | The benefits of an additional worker are task-dependent: Assessing low-back injury risks during prefabricated (panelized) wall construction | 2012 | *Applied Ergonomics* |
| 89 | Low back injury risks during construction with prefabricated (panelized) walls: effects of task and design factors | 2012 | *Ergonomics* |
| 90 | Offsite production: A model for building down barriers A European construction industry perspective | 2011 | *Engineering, Construction and Architectural Management* |
| 91 | Demystifying the cost barriers to offsite construction in the UK | 2010 | *Construction Management and Economics* |

## Appendix B

**Table A2.** List of 67 risk factors.

| Risk Factors | Code | Frequency | Weight | | | Total Weight (TW) | Total Score (TS) | Relative Score (RS) | Cumulative Score (CS) |
|---|---|---|---|---|---|---|---|---|---|
| | | | $W_h$ | $W_m$ | $W_l$ | | | | |
| Design and planning risks | | | | | | | | | |
| Change order/design freeze issues from the clients | DP-1 | 17 | 8 | 5 | 4 | 59 | 76 | 0.13 | 0.13 |
| Complexity in the modular designs/rigid geometry | DP-2 | 15 | 7 | 5 | 3 | 53 | 68 | 0.11 | 0.24 |
| Design changes and defects in the module size | DP-3 | 16 | 8 | 4 | 3 | 55 | 71 | 0.12 | 0.36 |
| Coordination problem between the project participants | DP-4 | 13 | 6 | 4 | 3 | 45 | 58 | 0.10 | 0.46 |
| Shop drawing management problems/unclarity | DP-5 | 10 | 5 | 3 | 2 | 36 | 46 | 0.08 | 0.53 |
| Lack of BIM and visualization techniques in the design | DP-6 | 8 | 4 | 2 | 2 | 28 | 36 | 0.06 | 0.59 |
| Inadequate codes and standards of the MiC locally | DP-7 | 11 | 6 | 3 | 2 | 41 | 52 | 0.09 | 0.68 |
| Delivery of shop drawings to the manufacturing plant | DP-8 | 5 | 3 | 1 | 1 | 19 | 24 | 0.04 | 0.72 |
| Inefficiency in design toward fire and seismic rules | DP-9 | 3 | 1 | 2 | | 11 | 14 | 0.02 | 0.74 |
| Superfluous activities during design | DP-10 | 4 | 2 | 1 | 1 | 14 | 18 | 0.03 | 0.77 |
| Inadequacy in adopting local codes | DP-11 | 6 | 3 | 2 | 1 | 22 | 28 | 0.05 | 0.82 |
| Errors and mistakes in the shop drawings | DP-12 | 5 | 3 | 2 | | 21 | 26 | 0.04 | 0.86 |
| Superfluous use of materials in design | DP-13 | 3 | 1 | 1 | 1 | 9 | 12 | 0.02 | 0.88 |
| Inadequacy in the drawing specification | DP-14 | 5 | 2 | 2 | 1 | 17 | 22 | 0.04 | 0.92 |
| Low adoption of sustainable and energy-efficient practices | DP-15 | 3 | 2 | 1 | | 13 | 16 | 0.03 | 0.95 |
| Insufficient brief of the design from the client's side | DP-16 | 4 | 3 | 1 | | 18 | 22 | 0.04 | 0.98 |
| Low consideration toward adjacent forces in the structure | DP-17 | 2 | 1 | 1 | | 8 | 10 | 0.02 | 1.00 |
| Offsite manufacturing risks | | | | | | | | | |
| Poor understanding of process plans/system failure | OM-1 | 16 | 9 | 4 | 3 | 60 | 76 | 0.11 | 0.11 |
| Noise, fume, and toxic compound exposure at the plant | OM-2 | 10 | 5 | 4 | 1 | 38 | 48 | 0.07 | 0.18 |
| Conflicts in geometry of modules from the design phase | OM-3 | 15 | 8 | 5 | 2 | 57 | 72 | 0.10 | 0.28 |
| Inadequate inventory control and shortage of material | OM-4 | 13 | 7 | 4 | 2 | 49 | 62 | 0.09 | 0.37 |
| Poor/inexperienced labor and resource allocation | OM-5 | 12 | 6 | 4 | 2 | 44 | 56 | 0.08 | 0.45 |
| Lack of modern equipment for lifting processes at the plant | OM-6 | 11 | 5 | 3 | 3 | 37 | 48 | 0.07 | 0.52 |
| Defects due to welding process/geometric variations | OM-7 | 10 | 6 | 2 | 2 | 38 | 48 | 0.07 | 0.58 |
| Inadequacy in weather proofing and space usage | OM-8 | 11 | 6 | 3 | 2 | 41 | 52 | 0.07 | 0.66 |
| Excessive production of modules due to information gap | OM-9 | 8 | 4 | 2 | 2 | 28 | 36 | 0.05 | 0.71 |
| Additional lead time for module production | OM-10 | 8 | 4 | 3 | 1 | 30 | 38 | 0.05 | 0.76 |
| Time lapsed during production inspections and approvals | OM-11 | 6 | 4 | 1 | | 23 | 29 | 0.04 | 0.81 |
| Disorganized verification process of modules | OM-12 | 6 | 3 | 2 | 1 | 22 | 28 | 0.04 | 0.85 |
| Poor integration of mechanical, electrical, and plumbing fit | OM-13 | 4 | 3 | 1 | | 18 | 22 | 0.03 | 0.88 |
| Inefficient planning of usage of factory space and equipment | OM-14 | 3 | 2 | 1 | | 13 | 16 | 0.02 | 0.90 |
| Insufficient production line conditions at the plant | OM-15 | 4 | 2 | 1 | 1 | 14 | 18 | 0.03 | 0.93 |
| Variations in production operation rate of modules | OM-16 | 5 | 2 | 2 | 1 | 17 | 22 | 0.03 | 0.96 |
| Low capacity of manufacturers and suppliers | OM-17 | 3 | 1 | 2 | | 11 | 14 | 0.02 | 0.98 |
| Diminished supply of quality and type of material | OM-18 | 3 | 2 | 1 | | 13 | 16 | 0.02 | 1.00 |

**Table A2.** *Cont.*

| Risk Factors | Code | Frequency | Weight | | | Total Weight (TW) | Total Score (TS) | Relative Score (RS) | Cumulative Score (CS) |
|---|---|---|---|---|---|---|---|---|---|
| | | | $W_h$ | $W_m$ | $W_l$ | | | | |
| Transportation and Logistics risks | | | | | | | | | |
| Delay in delivery/poor scheduling of modules | TL-1 | 13 | 7 | 4 | 2 | 49 | 62 | 0.11 | 0.11 |
| Defects by damage, flexing, warping and manual handling | TL-2 | 10 | 5 | 4 | 1 | 38 | 48 | 0.09 | 0.20 |
| Size and weight restrictions in transportation | TL-3 | 12 | 6 | 4 | 2 | 44 | 56 | 0.10 | 0.30 |
| Restrictions of rules, regulations, and transport vehicles | TL-4 | 11 | 5 | 3 | 3 | 37 | 48 | 0.09 | 0.38 |
| Early arrival and wrong delivery of modules onsite | TL-5 | 10 | 5 | 3 | 2 | 36 | 46 | 0.08 | 0.47 |
| Poor marking/tagging and improper buffer space onsite | TL-6 | 11 | 6 | 3 | 2 | 41 | 52 | 0.09 | 0.56 |
| Distance issues and taxes incurred between the plant and site | TL-7 | 8 | 4 | 2 | 2 | 28 | 36 | 0.06 | 0.62 |
| Misplacement of modules in warehouses cause delay | TL-8 | 8 | 4 | 3 | 1 | 30 | 38 | 0.07 | 0.69 |
| Inadequate availability of transportation vehicles | TL-9 | 6 | 4 | 2 | | 26 | 32 | 0.06 | 0.75 |
| Rate of freight issues during transportation | TL-10 | 6 | 4 | 1 | 1 | 24 | 30 | 0.05 | 0.80 |
| Worker error causing information gap | TL-11 | 5 | 3 | 1 | 1 | 19 | 24 | 0.04 | 0.85 |
| Transport route inefficiency reflecting the size of modules | TL-12 | 4 | 2 | 2 | | 16 | 20 | 0.04 | 0.88 |
| Extreme case of disruptions caused by weather | TL-13 | 4 | 2 | 1 | 1 | 14 | 18 | 0.03 | 0.91 |
| Congestion and traffic on roads cause delay | TL-14 | 5 | 2 | 2 | 1 | 17 | 22 | 0.04 | 0.95 |
| Inefficient vehicle and road conditions | TL-15 | 3 | 1 | 2 | | 11 | 14 | 0.03 | 0.98 |
| Accidents and unplanned happening of activities | TL-16 | 3 | 1 | 1 | 1 | 9 | 12 | 0.02 | 1.00 |
| Onsite assembly risks | | | | | | | | | |
| Inefficient lift path/layout planning of the crane(s) and scheduling/sequencing of modules | OA-1 | 16 | 8 | 5 | 3 | 58 | 74 | 0.11 | 0.11 |
| Poor stability/blind lifting, breakdown of the crane and frequent change in rigging direction | OA-2 | 15 | 7 | 4 | 4 | 51 | 66 | 0.10 | 0.20 |
| Break of the cable crane/jib falling and extra load on the crane at the site | OA-3 | 13 | 6 | 3 | 4 | 43 | 56 | 0.08 | 0.29 |
| Unsafe acts/conditions and error in installation onsite | OA-4 | 15 | 8 | 4 | 3 | 55 | 70 | 0.10 | 0.39 |
| Poor verification due to inadequate tagging and inefficient welding | OA-5 | 12 | 6 | 4 | 2 | 44 | 56 | 0.08 | 0.47 |
| Manual lifting, unwrapping, lining, unhooking, and screwing | OA-6 | 10 | 5 | 3 | 2 | 36 | 46 | 0.07 | 0.54 |
| Variabilities in geometry/dimensions and poor alignment of modules | OA-7 | 13 | 7 | 4 | 2 | 49 | 62 | 0.09 | 0.63 |
| Wind/weather and near environment disruptions at the site | OA-8 | 8 | 4 | 2 | 2 | 28 | 36 | 0.05 | 0.68 |
| Restrictions in site layout | OA-9 | 8 | 3 | 2 | 3 | 24 | 32 | 0.05 | 0.73 |
| Extra variation in foundation geometry | OA-10 | 7 | 4 | 2 | 1 | 27 | 34 | 0.05 | 0.78 |
| Overlapping of working space and radius of the crane | OA-11 | 6 | 4 | 1 | 1 | 24 | 30 | 0.04 | 0.82 |
| Inaccurate dimensioning of the modules | OA-12 | 6 | 4 | 2 | | 26 | 32 | 0.05 | 0.87 |
| Complex rectification of modules | OA-13 | 5 | 2 | 2 | 1 | 17 | 22 | 0.03 | 0.90 |
| Rework of the site layout due to a mismatch of drawings | OA-14 | 5 | 3 | 1 | 1 | 19 | 24 | 0.03 | 0.93 |
| Low experience of project managers onsite | OA-15 | 4 | 2 | 2 | | 16 | 20 | 0.03 | 0.96 |
| Incorrect inspection of modules upon arrival onsite | OA-16 | 5 | 3 | 2 | | 21 | 26 | 0.04 | 1.00 |

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
