# Peer review of "Volumetric Modular Construction Risks: A Comprehensive Review and Digital-Technology-Coupled Circular Mitigation Strategies"

_sustainability, doi:10.3390/su15087019_

Round 1
Reviewer 1 Report
The article presents a very pertinent theme, in the context of the crisis in the construction sector due to its environmental impact and the high labor prices that only industrialization can solve.
In general, the objectives, structure and methodology presented are adequate. I am of the opinion that the article would be greatly enriched if it resorted to practical cases that exemplified the concrete production, in terms of design and construction, within the scope of the VMC.
Author Response
Respected reviewer,
Please find attached the document stating the addressed comments.
Regards,
Author(s) of the manuscript

Reviewer 2 Report
Manuscript ID: sustainability-2302551
Type: review
Title: Volumetric Modular Construction: A review on risks and digital technology coupled circular mitigation strategies
The topic is meaningful, and my review comments on this interesting review are as follows.
1. The title of this article contains too much information. Although this can include more popular keywords and attract a wider audience, it cannot focus on the core research issues. I think the main focus of this review is to identify and summarize the CRFs of VMC. That is, the first two main research questions mentioned in Line 101. This can also be reflected from the search query rules of Line 140-145.
However, in the title, risk and DT are regarded as a parallel relationship, and its core and focus are shifted to "circular mitigation strategies", but I do not deny that, the discussion of Section 5 Mitigation framework for using digital technology-based circular strategies to overcome VMC risks, is very meaningful. If DT and CRFs are to be juxtaposed, it is obvious that a more systematic and comprehensive discussion on the classification and development of DT is required. This can be the work content of another article, and the workload of this article is sufficient.
2. The basis of the discussion is the selected 91 papers after a systematic literature review. The author’s handling process in the review part is standard. There is a small problem: Line 46. Regarding the query period, the author describes it as (2010- present), in order to let readers know the time range of the search more clearly, it is recommended that the author express the time more accurately. For example, “2020” is “January 1, 2020”? also, can the “present” provide a note to clarify the specific time of the research?
3. The resolution of Figure 5 is too low to see the text information and the relationship between them clearly.
4. Table 3 is not the author's results information, and it is of little significance to display it in a table. It is mainly the abbreviation and full name of the specific content of DFx. This content could be directly displayed in text at line 328.
5. Appendices A and B were not found in the manuscript.
Author Response

(The authors gave the same response as above.)

Reviewer 3 Report
Some corrections and modifications are necessary to improve this paper as follows:
1. What is the main question addressed by the research? The main question concerns the description of critical risk factors in Volumetric Modular Construction. As I could understand the concept of the research, it classifies the risks in VMC taking into account the project stages. It also examines the application of circular strategies concerning VMC. The manuscript highlights the potential of digital technologies based circular strategies.
2. Do you consider the topic original or relevant in the field? Does it address a specific gap in the field? In my opinion the topic is original. Previous studies identified different risks concerning cost and schedule performance, causal relationships, adoption and process barriers. This manuscript reviels new aspects.
3. What does it add to the subject area compared with other published material? As it was examined there are no studies that holistically addressed risks in VMC regarding the project stages and project attributes. From this point of view the description of strategies for reducing and mitigating risks in Figure 10 has relevance.
4. What specific improvements should the authors consider regarding the methodology? What further controls should be considered? The study highlights the potential of digital technologies based circular strategies to address the risks in VMC, however a quantitative assessment and validation is needed.
5. Are the conclusions consistent with the evidence and arguments presented and do they address the main question posed? Concerning the main question posed in the manuscript, it delivers a checklist of critical risk factors in the VMC stages and guides towards understanding those risks in detail. The study presents the potential of digital technology-based circular strategies to overcome the VMC risks that can be useful for various stakeholders of the VMC industry.
6. Are the references appropriate? The manuscript contains 127 references, which are in unified format. Titles are about modular, risk, industrialized and prefabrication corresponding to the main topic.
-Line 194: refer to Hosseini et al. 2018 without mentioning the year
-Line 241: space between 31 and articles
-Line 276: Figure 5 is not so clear
-Line 700: Figure 10 is not so clear
Author Response

(The authors gave the same response as above.)

Round 2
Reviewer 2 Report
I think these changes will make a strong impact on what the potential readers take from this paper.
I recommend this paper be accepted in its present form.